# Changing course: Glucose starvation drives nuclear accumulation of Hexokinase 2 in *S. cerevisiae*

Mitchell A. Lesko[1], Dakshayini G. Chandrashekarappa[2], Eric M. Jordahl[1], Katherine G. Oppenheimer[1], Ray W. Bowman II[1], Chaowei Shang[1], Jacob D. Durrant[1], Martin C. Schmidt[2]*, Allyson F. O'Donnell[1]*

1 Department of Biological Sciences, University of Pittsburgh, Pittsburgh, Pennsylvania, United States of America, 2 Department of Microbiology and Molecular Genetics, University of Pittsburgh School of Medicine, Pittsburgh, Pennsylvania, United States of America

* mcs2@pitt.edu (MCS); allyod@pitt.edu (AFO)

**Data Availability Statement:** The authors confirm that all data underlying the findings are fully available without restriction. All RNA-seq data have

## Abstract

Glucose is the preferred carbon source for most eukaryotes, and the first step in its metabolism is phosphorylation to glucose-6-phosphate. This reaction is catalyzed by hexokinases or glucokinases. The yeast *Saccharomyces cerevisiae* encodes three such enzymes, Hxk1, Hxk2, and Glk1. In yeast and mammals, some isoforms of this enzyme are found in the nucleus, suggesting a possible moonlighting function beyond glucose phosphorylation. In contrast to mammalian hexokinases, yeast Hxk2 has been proposed to shuttle into the nucleus in glucose-replete conditions, where it reportedly moonlights as part of a glucose-repressive transcriptional complex. To achieve its role in glucose repression, Hxk2 reportedly binds the Mig1 transcriptional repressor, is dephosphorylated at serine 15 and requires an N-terminal nuclear localization sequence (NLS). We used high-resolution, quantitative, fluorescent microscopy of live cells to determine the conditions, residues, and regulatory proteins required for Hxk2 nuclear localization. Countering previous yeast studies, we find that Hxk2 is largely excluded from the nucleus under glucose-replete conditions but is retained in the nucleus under glucose-limiting conditions. We find that the Hxk2 N-terminus does not contain an NLS but instead is necessary for nuclear exclusion and regulating multimerization. Amino acid substitutions of the phosphorylated residue, serine 15, disrupt Hxk2 dimerization but have no effect on its glucose-regulated nuclear localization. Alanine substitution at nearby lysine 13 affects dimerization and maintenance of nuclear exclusion in glucose-replete conditions. Modeling and simulation provide insight into the molecular mechanisms of this regulation. In contrast to earlier studies, we find that the transcriptional repressor Mig1 and the protein kinase Snf1 have little effect on Hxk2 localization. Instead, the protein kinase Tda1 regulates Hxk2 localization. RNAseq analyses of the yeast transcriptome dispels the idea that Hxk2 moonlights as a transcriptional regulator of glucose repression, demonstrating that Hxk2 has a negligible role in transcriptional regulation in both glucose-replete and limiting conditions. Our studies define a new model of *cis*- and trans-acting regulators of Hxk2 dimerization and nuclear localization. Based on our data, the nuclear translocation of Hxk2 in yeast occurs in glucose starvation conditions, which aligns

been deposited in the SRA database under accession number PRJNA885127.

**Funding:** This research was funded by National Science Foundation CAREER grant MCB 155143 and 1902859 to A.F.O, National Institutes of Health T32 grant GM133353 to M.A.L., R01 grant GM132353 to J.D.D., and R01 grant GM046443 to M.C.S. The funders had no role in the study design, data collection and analysis, decision to publish, or preparation of the manuscript.

**Competing interests:** The authors have declared that no competing interests exist.

well with the nuclear regulation of mammalian orthologs. Our results lay the foundation for future studies of Hxk2 nuclear activity.

## Author summary

Glucose is converted to energy in most cells. To initiate this conversion, enzymes called hexokinases must modify glucose, making them critical metabolic regulators. Mutations that change hexokinase function are associated with disease, including cancers and metabolic disorders. In addition to modifying glucose, which happens in the cytosol, hexokinases can move into the nucleus where their function is not well understood. We demonstrate that in yeast, hexokinase 2 moves into the nucleus when cells are starved for glucose and is nuclear excluded in rich glucose conditions. We define key regulators, both within hexokinase 2 itself and in hexokinase 2 interacting proteins, that control its nuclear localization. Finally, we demonstrate that hexokinase 2 does not directly control gene expression in response to changing glucose environments. Our work contradicts a long-standing view for hexokinase 2 nuclear regulation and function and presents a new model for how hexokinase 2 nuclear localization is contolled.

## Introduction

The functional complexity of proteomes is extended by "moonlighting" proteins, a term describing proteins with "other jobs" in the cell [1]. Enzymes with roles in signal transduction, independent of their catalytic activities, are good examples. For instance, cytochrome c is part of the mitochondrial electron transport chain. However, when released from the mitochondria, cytochrome c becomes a messenger of apoptotic signaling [2]. Glycolytic enzymes in many species have moonlighting functions distinct from their catalytic potential [3]. We examine the yeast hexokinase 2 (Hxk2), the enzyme that catalyzes the first step of glycolysis. Like other glycolytic enzymes, Hxk2 can translocate to the nucleus, suggesting that it may carry out a 'moonlighting' nuclear function distinct from its glycolytic role. Here we assess Hxk2 nuclear translocation and its potential nuclear function as a transcription regulator.

The budding yeast *Saccharomyces cerevisiae* express three enzymes capable of phosphorylating glucose, any one of which can support growth on glucose [4]. Two of these enzymes, Hxk1 and Hxk2, are hexokinases with broad substrate specificity, including glucose and fructose [5]. The third enzyme, Glk1, is a glucokinase, named for its glucose specificity. Hxk1 and Hxk2 are closely related paralogs with 77% identity and 89% similarity in their amino acid sequences. Glk1 is less closely related to the hexokinases (37% identity and 53% similarity) but has a paralog, Emi2, whose function is uncertain. Recombinant Emi2 has detectable glucose phosphorylating activity [6] but the presence of Emi2 is not sufficient to confer growth on glucose in *hxk1Δ hxk2Δ glk1Δ* cells, suggesting it may not function as a hexokinase *in vivo* [4].

Hxk2 is proposed to have a moonlighting function in the nucleus regulating gene expression [7–16]. The current model for Hxk2, proposed by the Moreno lab, states that Hxk2 translocates to the nucleus in glucose-rich conditions [15] in a manner dependent upon (1) binding to Mig1, a transcriptional repressor [8], (2) a nuclear localization signal (NLS) in the N-terminus of Hxk2 between lysine 6 and lysine 13 [14], and (3) dephosphorylation of Hxk2 at serine 15 [11]. Nuclear Hxk2 is proposed to be one subunit of a transcriptional repressor complex that includes the DNA binding proteins Mig1 and Mig2; the Tup1 repressor; Med8, a subunit of the Mediator complex; Reg1, a regulatory subunit of the PP1 phosphatase; and the nuclear

isoform of yeast AMP-activated protein kinase (AMPK) composed of the Snf1, Snf4, and Gal83 proteins. In this model, this large complex is needed for glucose repression of gene expression [7].

While the studies that support this model are cited collectively >1000 times [17–22], there has been no independent corroboration for the model, and aspects have been questioned in published commentaries [23]. The idea that yeast Hxk2 is excluded from the nucleus in glucose-starvation conditions and found in the nucleus in glucose-replete conditions is counter to many models of hexokinase/glucokinase regulation in other organisms [24–27]. Mammalian hexokinase isoforms II and III and glucokinase (aka hexokinase IV) can each be nuclear [24,25,28]. Most data suggest that these enzymes are nuclear in response to glucose limitation or stress, which is the opposite of what has been reported for yeast Hxk2 [24,25,28].

Unfortunately, entwined with the model of Hxk2 nuclear-cytosolic shuttling is the idea that Hxk2's oligomeric state regulates nuclear partitioning. Rigorous *in vivo* biochemical analyses demonstrate that in glucose-grown cells, a balance of monomeric and dimeric Hxk2 exists [29]. In glucose-starved cells, Hxk2 is predominantly monomeric [29–33]. A key regulator of Hxk2 dimerization is serine 15, which is phosphorylated in glucose-starvation conditions to disrupt the dimer [29–33]. The Tda1 kinase is required for serine 15 phosphorylation, while Snf1 plays only a minor role [29]. In cells lacking Tda1, Hxk2 remains a dimer in both glucose replete and restricted conditions [29], suggesting that Tda1 controls Hxk2 monomer-dimer balance. Unlike the clear link between serine 15 phosphorylation and Hxk2 monomer-dimer balance, the proposition that serine 15 regulates Hxk2 nuclear translocation is poorly supported [23].

We use high-resolution quantitative fluorescent imaging, biochemical and genetic methods to study the nuclear localization, dimerization, and function of Hxk2. Our data contradict all aspects of the current Hxk2 nuclear localization model. We demonstrate that Hxk2 is excluded from the nucleus in glucose-replete conditions, the very time when it is proposed to operate in a glucose repression transcriptional complex. We find that yeast Hxk2 enters the nucleus, but only in glucose starvation conditions, which is in line with the starvation and stress induced nuclear translocation of mammalian hexokinases [24,25]. Our imaging studies differ from earlier work, [8,11,13,14] in that we maintain the appropriate glucose supply throughout live-cell imaging, ensuring representation of the cellular response in these conditions.

We further define *cis* and *trans* regulatory elements that control the starvation-induced nuclear accumulation of Hxk2. We identify Hxk2 lysine 13 as required for the glucose-regulated nuclear exclusion of Hxk2 and dimerization. In contrast to earlier studies, mutation of serine 15 did not alter glucose-regulated nuclear accumulation of Hxk2. In keeping with Kriegel lab studies, serine 15 mutants of Hxk2 altered the monomer-dimer balance in cells [30–33]. Thus, Hxk2 glucose-regulated nuclear translocation and dimerization can be uncoupled. The Tda1 kinase, and not Snf1 or Mig1, was needed for Hxk2 nuclear translocation in response to glucose starvation.

Finally, our RNAseq analyses showed that the expression of the most highly glucose-repressed genes was not affected by loss of the *HXK2* gene. Taken together, our data refute the idea that Hxk2 moonlights as a transcriptional repressor important for glucose repression. The function of nuclear Hxk2 in glucose-starved cells remains to be defined.

## Results

### Hxk2 nuclear shuttling

We examined the impact of glucose abundance on the nuclear propensity of the three hexokinases in *S. cerevisiae*: Hxk1, Hxk2, and Glk1. Earlier studies suggested that Hxk2 from *S. cerevisiae* and *C. albicans* is nuclear excluded in glucose starvation and partially nuclear localized in

abundant glucose [8,11,13,14,34]. However, in these earlier studies, cells were incubated in glucose-free medium before imaging (glycerol-containing medium or PBS in the *S. cerevisiae* or *C. albicans* experiments, respectively), perhaps to allow for DAPI nuclear staining [8,11,13,14,34]. We generated functional GFP-tagged, plasmid-borne versions of these hexokinases and expressed them in cells lacking their respective endogenous genes. To examine the localization of these hexokinase-GFP fusions in glucose-replete conditions (2% glucose, "high" glucose) and after acute glucose starvation (0.05% glucose, "low" glucose), we used live-cell confocal microscopy and a mScarlet-tagged nuclear marker to quantify nuclear co-localization. In contrast to past qualitative Hxk2 studies [8,11,13,14], Hxk2 was largely cytosolic in glucose-grown cells, and a portion of Hxk2 became nuclear upon glucose starvation (Fig 1A).

We next quantified for each hexokinase: (1) the mean whole-cell fluorescence, (2) mean nuclear fluorescence, and (3) the mean nuclear to mean whole-cell fluorescence ratio to assess the relative contribution of the nuclear signal to that of total cellular fluorescence (Fig 1B–1D). This last measure is essential as it accounts for changes in gene expression or protein abundance/stability that accompany nutrient changes, as is evident for Hxk1 and Glk1 (Fig 1A–1D). In addition, higher whole-cell fluorescence can increase fluorescence in the nuclear compartment because background fluorescence increases. The ratio of the nuclear to whole cell fluorescence is thus the most useful measure to assess nuclear distribution. Manual and automated image analyses were performed in four biological replicate experiments. Data from manual and automated quantification were consistent, validating the automated quantification pipeline (Figs 1C, 1D, S1A and S1B). In response to glucose starvation, the mean nuclear fluorescence of Hxk2-GFP increased 3-fold, and the ratio of nuclear-to-whole-cell Hxk2-GFP fluorescence more than doubled, demonstrating a shift in Hxk2 to the nucleus upon glucose restriction. We also observed a glucose-starvation-induced increase in nuclear fluorescence with chromosomally-integrated, mNeonGreen-tagged Hxk2 (S1C–S1E Fig).

These data run counter to earlier findings that suggested Hxk2-GFP was retained in the nucleus in glucose-grown cells and excluded from the nucleus upon glucose starvation [8,11,13,14]. Three factors may account for this discrepancy. First, earlier studies incubated cells in glycerol or PBS (*i.e.*, lacking glucose) before imaging [8,11,13,14,34]. This pre-incubation could provoke the glucose starvation-induced nuclear translocation we observed. When we imaged *S. cerevisiae* after incubation in 80% glycerol, we found nuclear accumulations of Hxk2-GFP regardless of the medium in which cells were pre-grown (S1F Fig). Second, high-copy plasmids were used to express Hxk2-GFP in the earlier studies, so this overexpression might have made localization changes difficult to interpret. Third, our studies differ from earlier studies due to improvements in microscopy and the lack of quantification of earlier datasets [8,11,13,14].

## Hxk1 and Glk1 nuclear shuttling

In contrast to Hxk2, we saw no nuclear accumulation of Hxk1 in glucose-grown or -starved cells (Fig 1A). Hxk1 transcription is upregulated upon glucose starvation, resulting in an ~2.5-fold change in Hxk1 whole-cell fluorescence (Fig 1B). Considering only mean nuclear fluorescence, Hxk1 appears to undergo modest nuclear accumulation in glucose starvation, likely due to increased total Hxk1-GFP expression (Fig 1C). When data were normalized to account for protein abundance changes using the ratio of mean nuclear to mean whole-cell fluorescence, we saw no change in the relative distribution of Hxk1 upon glucose restriction (Fig 1C–1D). The abundance of Hxk1 was lower than that of Hxk2 in glucose-replete conditions, but in glucose starvation, Hxk1 levels rose higher than those of Hxk2 (Fig 1B). If there was a significant accumulation of Hxk1 in the nucleus, we should have been able to detect it in the glucose starvation conditions, and we did not (Fig 1A–1C).

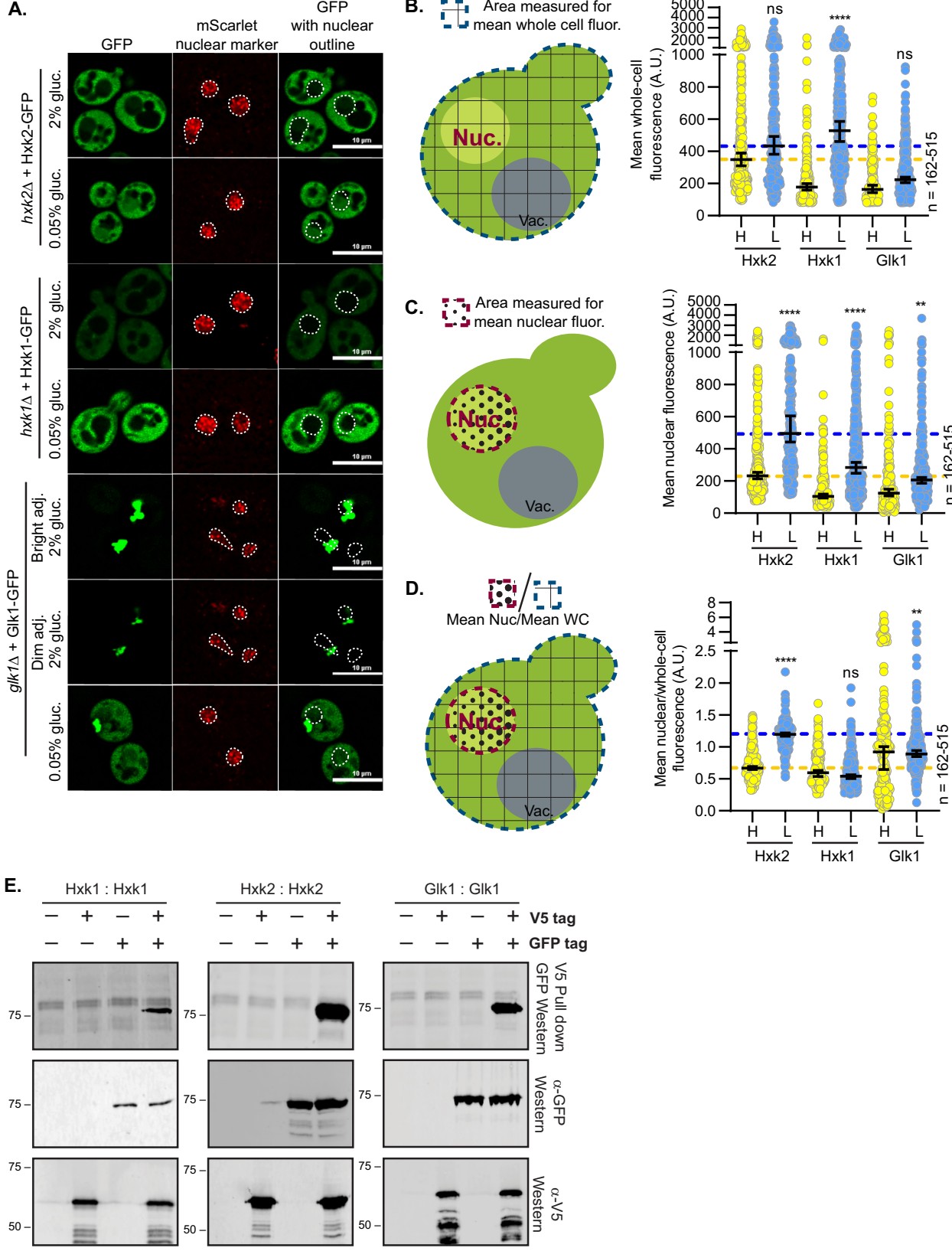

**Fig 1. Hexokinases alter localization in response to glucose starvation and can form multimers.** (A) Confocal microscopy of GFP-tagged yeast hexokinases expressed from CEN plasmids under the control of their endogenous promoters. Each hexokinase is expressed in cells where the endogenous gene has been deleted. Co-localization with the nucleus is determined based on overlap with the Tpa1-mScarlet nuclear marker, and a dashed white line indicates the nucleus. (B-D) Automated quantification (using Nikon.*ai* and GA3 analyses) of the images shown in panel A where (B) shows the mean whole-cell fluorescence intensities, (C) shows the mean nuclear fluorescence, (D) shows the mean nuclear fluorescence over the whole-cell fluorescence for each hexokinase as XY-scatter plots. The regions measured for each of these analyses are shown to the left of each graph with (B and D) the area for whole-cell measurement shown as a blue-dashed line filled with grid lines and (C-D) the area for the nuclear measurement shown as a red-dashed line filled with black dots. Horizontal bars show the median, and error bars indicate the 95% confidence interval. Dashed yellow and blue lines represent the median value for Hxk2-GFP cells in high and low glucose, respectively. Kruskal–Wallis statistical analysis with Dunn's post hoc test was performed to compare the (B) mean whole-cell fluorescence, (C) mean nuclear fluorescence, or (D) mean nuclear/whole-cell fluorescence ratio between high and low glucose medium conditions for each hexokinase. (E) To assess multimerization, we prepared extracts from yeast cells grown in 2% glucose and expressing the indicated hexokinase proteins either untagged (-) or tagged (+) with V5 or GFP. Protein inputs were monitored by immunoblotting (bottom two panels). Association of tagged proteins was assessed by co-immunoprecipitation using anti-V5 beads followed by immunoblotting with anti-GFP (top panel).

Consistent with recent reports [35], Glk1 formed cytosolic inclusions in high glucose (Fig 1A). These inclusions were diminished in low-glucose conditions, where Glk1 may be mobilized to help phosphorylate glucose [35]. Our Glk1 quantitative analyses were somewhat confounded by the large cytosolic inclusions often present near the nuclear marker; some cytosolic fluorescent signal was captured in the nucleus due to this overlap. Glk1 is transcriptionally upregulated in response to glucose starvation, causing a modest increase in its whole-cell fluorescence intensity (Fig 1B) [36]. As with Hxk1, changes in mean nuclear fluorescence suggested Glk1 may modestly accumulate in the nucleus in glucose starvation, but when normalized to whole-cell-fluorescence intensity, we observed no change in the relative distribution of Glk1 upon glucose restriction (Fig 1C–1D). These findings suggest that while Hxk2 undergoes a glucose starvation-induced nuclear translocation, Hxk1 and Glk1 do not.

## Hexokinase multimerization

Hxk2 nuclear localization could be linked to its transition from a dimer to a monomer. In cells grown in glucose-replete conditions, Hxk2 exists in a balance between dimeric and monomeric species [29,33]. Upon glucose starvation of cells, this balance shifts toward the monomeric state [29,33]. Hxk2 phosphorylation at S15 by the Tda1 kinase is vital for the transition to the monomer [29]. The Moreno lab suggested that Hxk2 is both nuclear and cytosolic in rich glucose conditions, but in response to glucose starvation, Hxk2 reportedly becomes nuclear excluded [8,11,13,14]. Further, this same lab showed that the phosphomimetic S15D mutation produced a nuclear-excluded Hxk2. Since S15D also gives rise to monomeric Hxk2, they argued that monomeric Hxk2 might be nuclear excluded [11], an idea contested in the literature [23].

Given the uncertainty of the relationship between Hxk2 monomer-dimer regulation and nuclear propensity, we assessed *in vivo* Hxk2 multimerization, as well as multimerization of Hxk1 and Glk1. We performed co-purification assays from yeast cells expressing differentially tagged forms of either Hxk1, Hxk2, or Glk1. In all cases, the V5-purified versions co-purified with the GFP-tagged Hxk2, providing evidence for *in vivo* interaction (*i.e.*, the formation of a dimer or higher-order multimer) (Fig 1E). We found evidence for heterodimer or multimer formation between Hxk1 and Hxk2 (S1G Fig), which co-purify in high-content yeast studies [37]. Given that all three hexokinases can multimerize, yet Hxk1 and Glk1 do not undergo nuclear translocation, multimerization is unlikely to explain the differences in their nuclear propensities.

## Dynamics of Hxk2 nuclear partitioning in response to glucose starvation

To ensure that Hxk2-GFP was functional, we assessed the ability of Hxk2-GFP and an untagged Hxk2 to support growth on glucose when present in *hxk1Δ hxk2Δ glk1Δ* cells and found that either of these forms supported robust growth (S2A Fig). We examined Hxk2-GFP nuclear partitioning in response to glucose starvation by performing a time-course image

analysis. In low glucose, we observed increased nuclear Hxk2-GFP within 15 minutes, with nuclear accumulation further increasing at 4 and 8 hours before declining at the 24-hour mark (Figs 2A, 2C and S2B–S2D). In prolonged starvation, nuclear accumulation of Hxk2-GFP diminished, likely due to increased protein degradation as evidenced by the accumulation of free-GFP that begins at 8 h post starvation and increases after 24 h of starvation (S2B–S2E Fig). Note that there is almost no free-GFP at early timepoints in glucose starved cells (1–4 h) (S2E Fig).

To determine the exchange rate between nuclear and cytosolic Hxk2, we performed fluorescence recovery after photobleaching (FRAP). We bleached the nuclear Hxk2-GFP in glucose-starved cells and monitored recovery of nuclear signal. In the 20 min post-bleaching, there was modest recovery of nuclear signal (~15% of its initial Hxk2-GFP nuclear signal), suggesting a slow exchange between nuclear and cytosolic Hxk2 (Figs 2D and S2F and S1 Movie). Nuclear Hxk2-GFP recovered only 1.4% of its fluorescence per minute (Fig 2D).

As a control, we monitored nuclear recovery in cells expressing free-GFP. Free-GFP transitioned between the nucleus and cytosol [38], recovering nuclear fluorescence at a rate of ~14%/minute and plateauing at 60% of original nuclear signal (Figs 2D and S2F). Unlike GFP, which freely diffuses into the nucleus, Hxk2-GFP has a slow, regulated nuclear translocation.

## Modification of Hxk2 at serine 15 does not alter nuclear partitioning but does prevent dimerization

Hxk2 phosphorylation at S15 regulates the dimer-to-monomer transition [31–33]. This site is sometimes called S14 because proteolytic processing removes the Hxk2 N-terminal methionine [39]. Others have reported that in glucose-starvation, phosphorylation at S15 results in nuclear-excluded Hxk2 [11]. We found that Hxk2 with S15 mutated to alanine or the phospho-mimetic aspartic acid (Hxk2$^{S15A}$ and Hxk2$^{S15D}$, respectively) had equivalent nuclear fluorescence to wild-type Hxk2 in glucose-replete and -starvation conditions (Fig 3A–3B). Regulation of Hxk2$^{S15A}$ and Hxk2$^{S15D}$ nuclear translocation appeared the same as wild-type Hxk2, occurring upon glucose restriction (Fig 3A–3C). These results suggest that Hxk2 nuclear propensity does not depend on S15, contradicting earlier studies [11].

We considered the possibility that the role of S15 in Hxk2 localization could be strain specific. Our studies used BY4741-derived yeast, related to the S288C background [40], but earlier studies used W303-derived yeast [8,11,13,14]. We repeated our experiments in W303-derived cells and observed similar Hxk2 nuclear regulation to what we found in BY4741 (S3A–S3B Fig). Based on these data, S15 does not alter Hxk2 nuclear regulation.

Hxk2-S15 mutant protein abundances and ability to support growth on glucose in *hxk1Δ hxk2Δ glk1Δ* cells were indistinguishable from wild-type Hxk2, demonstrating that they encode stable and active hexokinases (Fig 3D–3E). When we assessed *in vivo* multimerization, we found that Hxk2$^{S15D}$ failed to multimerize and Hxk2$^{S15A}$ reduced multimerization, suggesting that each mutant diminished Hxk2 dimer abundance (Fig 3F). Hxk2$^{S15D}$ disrupts Hxk2 dimerization *in vivo and in vitro*, while Hxk2$^{S15A}$ has a more modest impact on dimerization *in vitro* [29,31–33]. Consistent with our observations (Fig 3E), Hxk2$^{S15D}$ and Hxk2$^{S15A}$ have identical catalytic activities *in vitro* [31–33]. These data confirm that S15 is critical for Hxk2 multimerization, but changes to S15 did not alter Hxk2 nuclear propensity in wild-type cells. Nuclear translocation was still regulated by glucose starvation.

## Hxk2 homology modeling provides insight into the molecular mechanism of dimerization

To understand the mechanisms that govern Hxk2 dimerization and nuclear translocation, we generated a homology model of dimeric *S. cerevisiae* Hxk2 (referred as *Sc*Hxk2) based on a

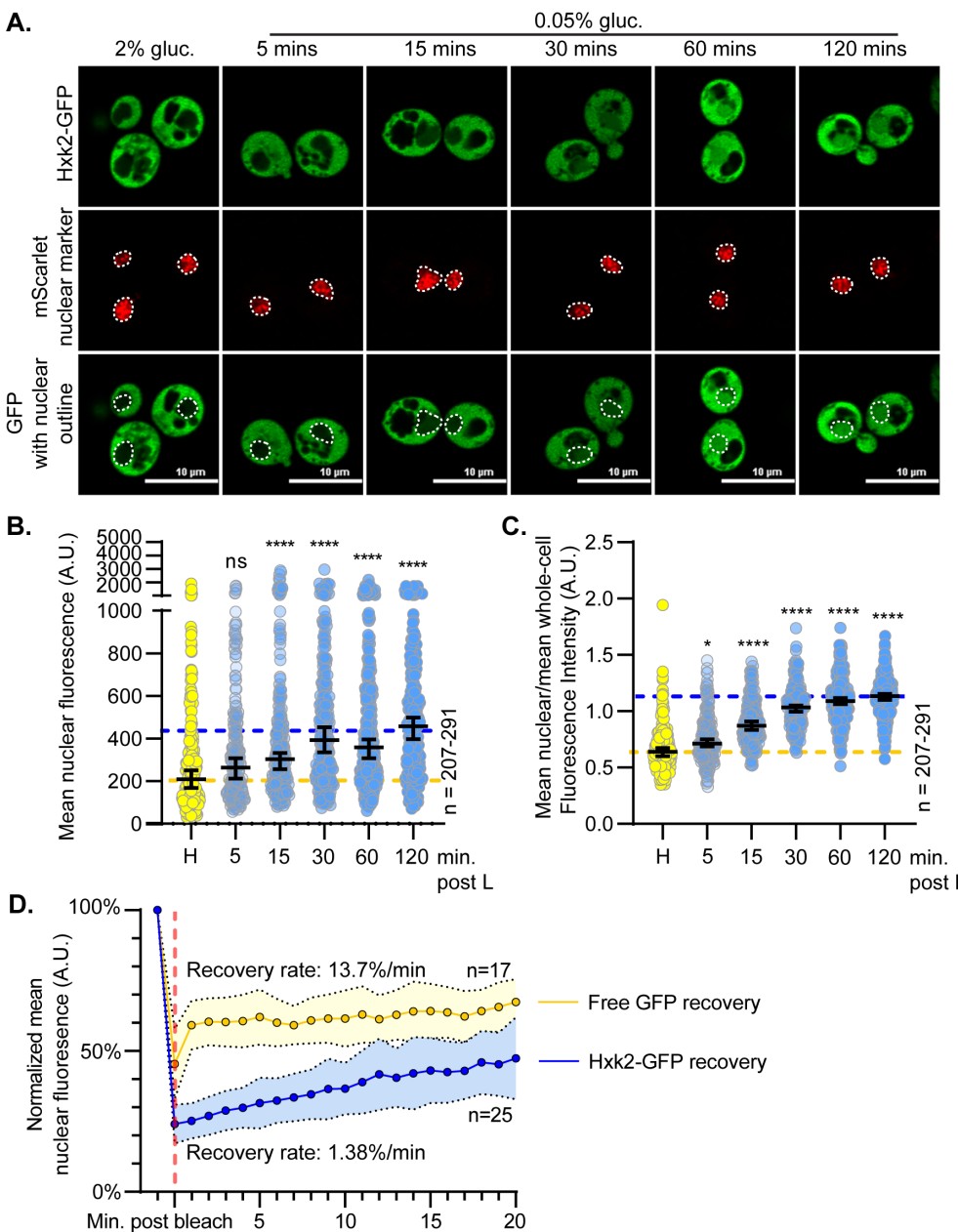

**Fig 2. Hxk2 increases its nuclear propensity upon shift to glucose starvation conditions.** (A) Confocal microscopy of GFP-tagged Hxk2 expressed from a CEN plasmid under the control of its own promoter in *hxk2Δ* cells. Co-localization with the nucleus is determined based on overlap with the Tpa1-mScarlet nuclear marker, and a dashed white line indicates the nucleus. (B-C) Automated quantification of images shown in panel A to measure (B) mean nuclear fluorescence or (C) the ratio of the mean nuclear/whole-cell fluorescence. Horizontal black lines show the median, and error bars indicate the 95% confidence interval. Dashed yellow and blue lines represent the median value for Hxk2-GFP in high and low glucose, respectively. Kruskal-Wallis statistical analysis with Dunn's post hoc test was performed to determine if the values obtained post low-glucose shift were statistically different from those obtained in high-glucose conditions. (D) Quantification of FRAP experiments done with cells expressing GFP-tagged Hxk2 from CEN plasmids under the control of its own promoter, and free GFP from CEN plasmids under the control of the *TEF1* promoter. The post-bleaching recovery rate was calculated by measuring the slope of the linear portion of each graph. Blue and gold dots represent the percentage of nuclear fluorescence recovered for Hxk2-GFP and free GFP, respectively. The vertical, red dashed line represents the time point at which nuclear ROI bleaching occurred.

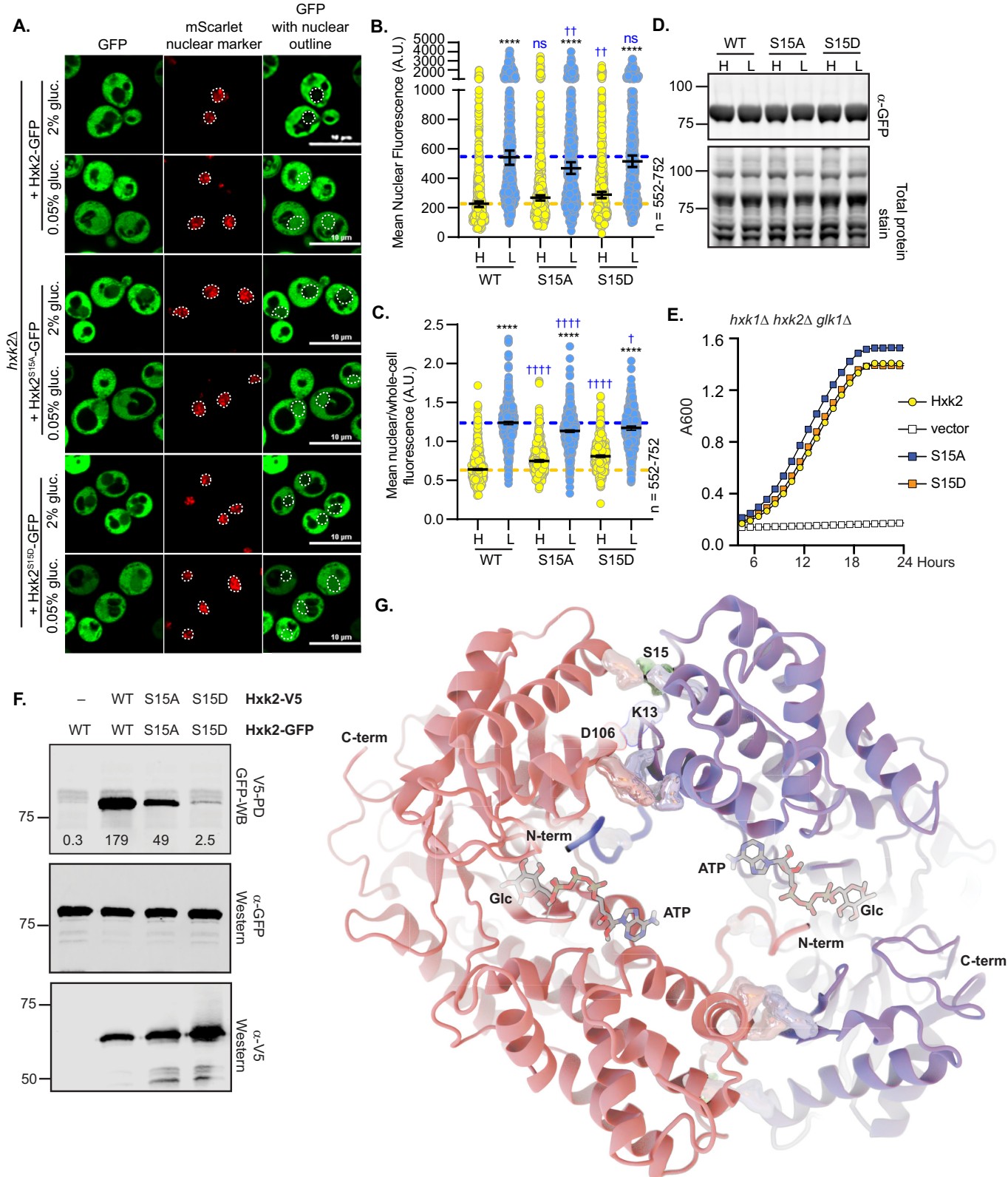

**Fig 3. Mutation of Hxk2 at S15 does not alter the regulation of its nuclear translocation but does change its ability to form multimers.** (A) Confocal microscopy of GFP-tagged Hxk2 or mutant forms, expressed from a CEN plasmid under the control of the *HXK2* promoter in *hxk2Δ* cells. Co-localization with the

nucleus is determined based on overlap with the Tpa1-mScarlet nuclear marker, and a dashed white line indicates the nucleus. (B-C) Automated quantification of the images shown in panel A to measure (B) mean nuclear fluorescence or (C) the ratio of the mean nuclear/whole-cell fluorescence. Horizontal black lines show the median, and error bars indicate the 95% confidence interval. Dashed yellow and blue lines represent the median value for Hxk2-GFP in high and low glucose, respectively. Black asterisks represent statistical comparisons between low and high glucose for a specific *HXK2* allele, and blue daggers represent statistical comparisons between mutant alleles and the corresponding WT Hxk2 in the same medium condition. (D) Immunoblot analyses of Hxk2-GFP from whole cell protein extracts made from cells grown in high glucose or shifted to low glucose for 2 hours. REVERT total protein stain serves as a loading control. (E) Cells lacking all three hexokinase genes (*hxk1Δ hxk2Δ glk1Δ*) were transformed with empty vector or plasmids expressing wild-type Hxk2, Hxk2$^{S15A}$, or Hxk2$^{S15D}$. Cell growth (A$_{600}$) in media containing glucose as the carbon source was monitored for 24 hours. (F) Extracts were prepared from yeast cells expressing the Hxk2 tagged with either V5 or GFP. Hxk2 proteins contained wild-type (WT) S15 or the S15A or S15D mutations. Protein expression was monitored by immunoblotting (bottom two panels). The association of the tagged proteins was assessed by co-immunoprecipitation using anti-V5 beads followed by western blotting with anti-GFP (top panel). Quantitation of the signal in the top panel is shown. (G) The Hxk2-dimer model, with the two monomers shown as pink and blue ribbons, respectively. The glucose molecule and ATP molecules are shown as sticks. The N- and C-terminal tails are marked with "N-term" and "C-term," respectively. All residues predicted to participate in intermonomer electrostatic interactions are shown as pink and blue metallic surfaces per the associated monomer (see Table 1 for residue numbers). Residue S15 is shown as a metallic green surface.

crystal structure of the *K. lactis* Hxk1 dimer (PDB 3O1W; referred to as *Kl*Hxk1) [41]. In *K. lactis*, the sole Hxk1 ortholog corresponds to both *Sc*Hxk1 and *Sc*Hxk2. *Sc*Hxk1 and *Sc*Hxk2 are paralogs that arose from a whole genome duplication in *S. cerevisiae*, which did not occur in *K. lactis* [42,43]. *Kl*Hxk1 is like both *Sc*Hxk1 (70% identity) and *Sc*Hxk2 (73% identity) (S4A Fig), but it is slightly more similar to *Sc*Hxk2, as is evident when structures of these three enzymes are superimposed (S4B–S4D Fig) [41]. The similarity between *Kl*Hxk1 and *Sc*Hxk2 (S4C Fig) makes *Kl*Hxk1 ideal for modeling dimeric *Sc*Hxk2. Crystal structures of many proteins lack N-terminal tails, which are often disordered, but the 3O1W [41] structure covers almost all the *Kl*Hxk1 sequence without gaps, including the two N-terminal tails. Each 3O1W N-terminus extends into the enzymatic cleft of the opposite *Kl*Hxk1 monomer, which may lock it into a stable position that can be crystallographically resolved. Our *Sc*Hxk2 homology model is similarly complete, including the cleft-bound N-terminal tails (Fig 3G).

To identify molecular interactions responsible for dimerization, we used BINANA [44,45] to find inter-chain interactions in the modeled dimer (Table 1 and highlighted in Fig 3G). This analysis identified two primary regions. The first is the N-terminal tail itself. Several charged tail residues participate in salt bridges with the opposite monomer (V2-D417*, K7-E457*, K13-D106*, where an asterisk denotes a residue belonging to the opposite monomer). The Q10 sidechain also forms hydrogen bonds with T107* and F105*. The second region of inter-chain interactions is at the interface between the two distal lobes, where the two monomers meet. Here K111 forms salt bridges with two residues (E359* and D360*), as does R113 (E359*

**Table 1. Summary of residues at the Hxk2 dimer interface per the homology model of the *Sc*Hxk2 dimer.**

| Hxk2 Monomer 1 | Hxk2 Monomer 2 | Interaction type |
|:---:|:---:|:---:|
| VAL 2 | ASP 417 | Salt bridge |
| LYS 7 | GLU 457 | Salt bridge |
| GLN 10 | PHE 105 | Hydrogen bond |
| GLN 10 | THR 107 | Hydrogen bond |
| LYS 13 | ASP 106 | Salt bridge |
| GLN 109 | GLU 356 | Hydrogen bond |
| LYS 111 | ASN 357 | Hydrogen bond |
| LYS 111 | GLU 359 | Salt bridge |
| LYS 111 | ASP 360 | Salt bridge |
| ARG 113 | GLU 359 | Salt bridge |
| ARG 113 | ASP 363 | Salt bridge |
| GLU 141 | LYS 379 | Salt bridge |

and D363*). E141 forms a single salt bridge with K379*, and the Q109 and K111 backbones form hydrogen bonds with E356* and N357*, respectively.

Based on our analyses of the ScHxk2 dimer model, it is not surprising that the two S15 mutants impact dimerization, as S15 is close to two inter-chain salt bridges (E141-K379* and K13-D106*) (Table 1 and Fig 3G). S15 phosphorylation (-2 e charge) would change the regional electrostatics, possibly disrupting dimer-promoting interactions. Mutation of this site to Ala would preclude phosphorylation and preserve the interactions, leaving the dimer intact (Table 1 and Fig 3G).

The cleft-bound N-terminal tails appear to play an important role in promoting ScHxk2 dimerization (Figs 3G and 4A). Given the crystallographic positions of bound glucose and ATP observed in other structures (e.g., 6PDT [35]), there do not appear to be substantial steric clashes between these bound substrates and the N-terminal tail, even if all three were to occupy the same cleft. We posit that the bound tail might instead be incompatible with catalytic-cleft closure, as required for glucose phosphorylation. Indeed, our homology model and published crystal structures [41] suggest that hexokinase dimerization generally—and perhaps N-terminal-tail binding specifically—maintains the catalytic cleft in an open conformation. In high glucose concentrations, bound glucose might disrupt N-terminal-tail binding within the catalytic pocket to destabilize the dimer (Fig 4A), and this could help form the monomer-dimer balance of Hxk2 observed in glucose-grown cells [29]. Alternatively, N-terminal-tail binding may prevent glucose binding in low glucose concentrations, encouraging dimerization. However, in vivo the Tda1 kinase phosphorylates Hxk2, which helps preserve it as a monomer even in glucose starvation conditions. When the Tda1 kinase is lost, along with its accompanying Hxk2 posttranslational modifications, Hxk2 is retained as a dimer in low glucose conditions [29], as would be predicted from our model.

We tested this model using molecular dynamics simulations. We found evidence that a glucose molecule and the N-terminal tail may not simultaneously occupy the catalytic pocket of the dimer (see S5A–S5F Fig). However, in simulations of the dimer where glucose was omitted, the N-terminal tail remained stably associated throughout the simulation (see S5A–S5F Fig and S1 Text). This finding is consistent with the idea that the N-terminal tail and its modification are key in regulating the dimer-to-monomer transition.

To verify that glucose promotes dimer dissociation independently of S15 phosphorylation, we used chromatography to biochemically examine multimerization of purified Hxk2 and Hxk2$^{S15D}$. Using nickel affinity columns, we obtained highly purified Hxk2 and Hxk2$^{S15D}$ from E. coli (Fig 4B). As a control, we confirmed that Hxk2$^{S15D}$ promotes monomerization in the absence of glucose. In size exclusion chromatography, wild-type Hxk2 eluted as a single peak with an ~8.8 mL elution volume. Hxk2$^{S15D}$ took longer to elute from the column (elution volume of ~9.5 mL), consistent with a smaller size than wild-type Hxk2 (Fig 4C, compare yellow curves). This change in elution profile is consistent with wild-type Hxk2 forming a larger, dimeric complex than the Hxk2$^{S15D}$ mutant in the absence of glucose. In contrast, Hxk2$^{S15D}$ cannot dimerize unless much higher enzyme concentrations are used [32] and represents monomeric Hxk2 [31,32].

We next preincubated Hxk2 with glucose in the absence of ATP, locking the enzyme in a glucose-bound state before performing size exclusion chromatography. In the presence of glucose, Hxk2 migrated slowly and eluted as a single peak at ~9.5 mL, the same elution profile observed with monomeric Hxk2$^{S15D}$ (Fig 4C, compare blue curve for WT Hxk2 to yellow Hxk2$^{S15D}$ curve). Unlike wild-type Hxk2, the elution profile of the monomeric Hxk2$^{S15D}$ did not change upon adding glucose, as expected since it was already monomeric (Fig 4I, compare yellow and blue curves for Hxk2$^{S15D}$). Since these assays were done without ATP, unphosphorylated glucose remained in the binding pocket. They are thus comparable to the simulations, which similarly omitted ATP.

These results confirm that glucose binding disrupts dimer formation. They support earlier biochemical studies demonstrating that glucose binding (1) promotes Hxk2 monomer formation and (2) greatly decreases the dimerization association constant. The Hxk2 dimer had a $K_a$ of 1.2 x $10^7$/M when glucose was absent versus 4.5 x $10^4$/M in the presence of glucose. A similar mutant to the one used here, Hxk2$^{S15E}$, had a dimer $K_a$ of 1.3 x $10^4$/M without glucose present, nearly 1000-fold lower than wild-type Hxk2 in the same conditions [32]. We propose a model that incorporates our MD simulations and biochemical findings with the earlier work from the Kriegel lab: (1) Glucose and N-terminal tail binding in the Hxk2 catalytic cleft are incompatible. (2) N-terminal tail docking into the catalytic cleft of the opposing monomer stabilizes Hxk2 dimerization. (3) Modification of S15 disrupts the dimer interface by occluding the N-tail/catalytic-pocket association between opposing monomers.

## The Hxk2 N-terminal tail, thought initially to contain an NLS, is required for Hxk2 dimer formation and nuclear exclusion

To assess the role of the N-terminal tail, we made a mutant Hxk2 lacking amino acids 7–16 (referred to as Hxk2$^{\Delta7-16}$). The Moreno group has studied this same mutation. They referred

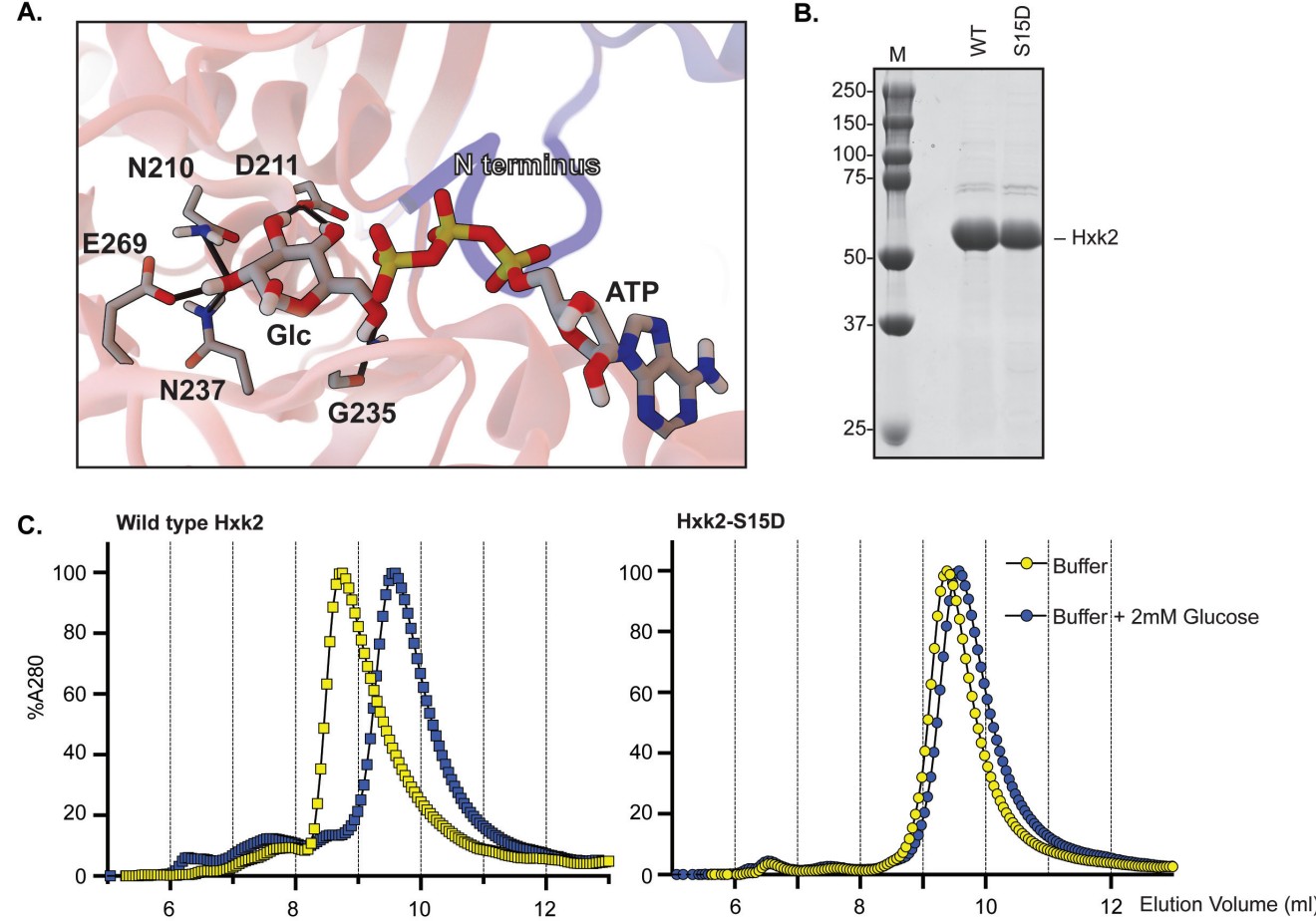

**Fig 4. Glucose binding prevents Hxk2 dimer formation.** (A) A close-up view of the enzymatic clefts of the Hxk2 dimer model (shown in Fig 3F). The N-terminus from one monomer (in blue) binds in the catalytic pocket of the opposing monomer (in pink). Positioned glucose and ATP molecules are shown in sticks representation. Hydrogen bonds are shown as solid black lines. (B) Yeast Hxk2 proteins (wild type and Hxk2-S15D) were expressed in and purified from *E. coli*. Each protein (5 μg) was resolved by SDS gel electrophoresis and stained with Coomassie blue. (C) Purified, recombinant Hxk2 proteins (WT, square symbols; Hxk2-S15D, round symbols) were resolved by size exclusion chromatography using a buffer with (blue) and without (yellow) glucose.

to it as either Hxk2$\Delta$K$^7$M$^{16}$, described as having lost its nuclear localization sequence [15], or the "without regulatory function" (WRF or Hxk2$^{WRF}$) mutation, described as having lost its ability to regulate glucose-repression of the *SUC2* gene [8,12]. Studies from the same lab suggest that Hxk2$^{\Delta7-16}$ retains full catalytic function but fails to bind Mig1 and does not localize to the nucleus, leading to their claim that amino acids in this region constitute an Hxk2 NLS [8,12].

We demonstrated that the Hxk2$^{\Delta7-16}$ mutant retains only partial catalytic activity (~50% that of WT Hxk2) and maintains glucose repression of the *SUC2* gene [4], refuting earlier claims [8,12]. Since this stretch of amino acids was reported as an NLS [15], we were surprised to find that Hxk2$^{\Delta7-16}$ localized to the nucleus in both glucose-replete and glucose-starvation conditions (Fig 5A–5C). In contrast to the reports that this region serves as an NLS [15], the Hxk2 N-terminus seems critical for maintaining glucose-regulated nuclear exclusion of Hxk2.

The Hxk2$^{\Delta7-16}$ protein was stable and functional, as confirmed by immunoblotting and its ability to support robust growth on glucose for cells lacking endogenous hexokinase (Fig 5D–5E). However, the Hxk2$^{\Delta7-16}$ mutant did not copurify a differentially tagged version of Hxk2$^{\Delta7-16}$, suggesting it cannot multimerize (Fig 5F). This result is not surprising considering our modeling and simulations, in which the Hxk2 N-terminal tail is necessary for dimerization (Fig 3G and Table 1).

Since residues 7–16 are not a bona fide NLS, we mapped surface-exposed lysines and arginines (S6A Fig) and used several NLS prediction tools [46–50] to try to identify the sequence responsible for Hxk2 nuclear translocation. Two of these tools identified a string of lysines at K54, K58, and K59 as a putative NLS. When we mutated these residues to alanine, the nuclear partitioning of Hxk2 in response to glucose fluctuations was unchanged (S6B–S6D Fig). Hxk2 does not contain an easily definable, canonical K/R-rich NLS.

## K13 in the Hxk2 N-terminal tail is required for Hxk2 dimer formation and glucose regulation of nuclear localization

There are three lysines among amino acids 7–16, and lysines are often critical for NLS function [51]. We made site-directed mutants at lysines 7, 8, and 13, converting these residues to alanine (referred to as Hxk2$^{K7,8,13A}$). This triple mutant recapitulated the high glucose nuclear localization observed for Hxk2$^{\Delta7-16}$ (S7A–S7C Fig). Rather than forming an NLS, this region helps maintain glucose-induced nuclear exclusion of Hxk2.

We found that the Hxk2$^{\Delta7-16}$ and Hxk2$^{K7,8,13A}$ mutants prevented dimerization and circumvented glucose-regulated nuclear translocation, giving rise to a pool of constitutively nuclear Hxk2. However, disrupting dimerization is not sufficient to drive nuclear localization of Hxk2 because Hxk2$^{S15D}$, which prevents dimerization, retains glucose-regulated nuclear exclusion (Figs 3A–3G, S3A and S3B). Amino acids 7–16 in Hxk2 are conserved in Hxk1, yet Hxk1 does not accumulate in the nucleus in glucose starvation (Fig 6A and Fig 1A–1D), and analogous mutations in Hxk1 to those under study for Hxk2 did not result in accumulation of nuclear Hxk1 (S8A–S8C Fig).

To refine the region responsible for glucose-regulated nuclear localization, we considered our *Sc*Hxk2 dimer homology model, which suggests the N-terminal-tail residues K7 and K13 form important salt bridge interactions that may stabilize the dimer (Figs 3G and 6B). Of these two sites, K13 may be dimethylated or sumoylated in Hxk2 [52,53] (Fig 6B) but ubiquitinated in Hxk1 [54]; this putative differential regulation could account for the distinct Hxk1 and Hxk2 nuclear localizations. We assessed the localization of GFP-tagged Hxk2$^{K13A}$, a mutation likely to disrupt the K13-D106 electrostatic interaction observed in the *Sc*Hxk2 model (Fig 6B, center panel). Like Hxk2$^{\Delta7-16}$ and Hxk2$^{K7,8,13A}$, Hxk2$^{K13A}$ was nuclear localized in both

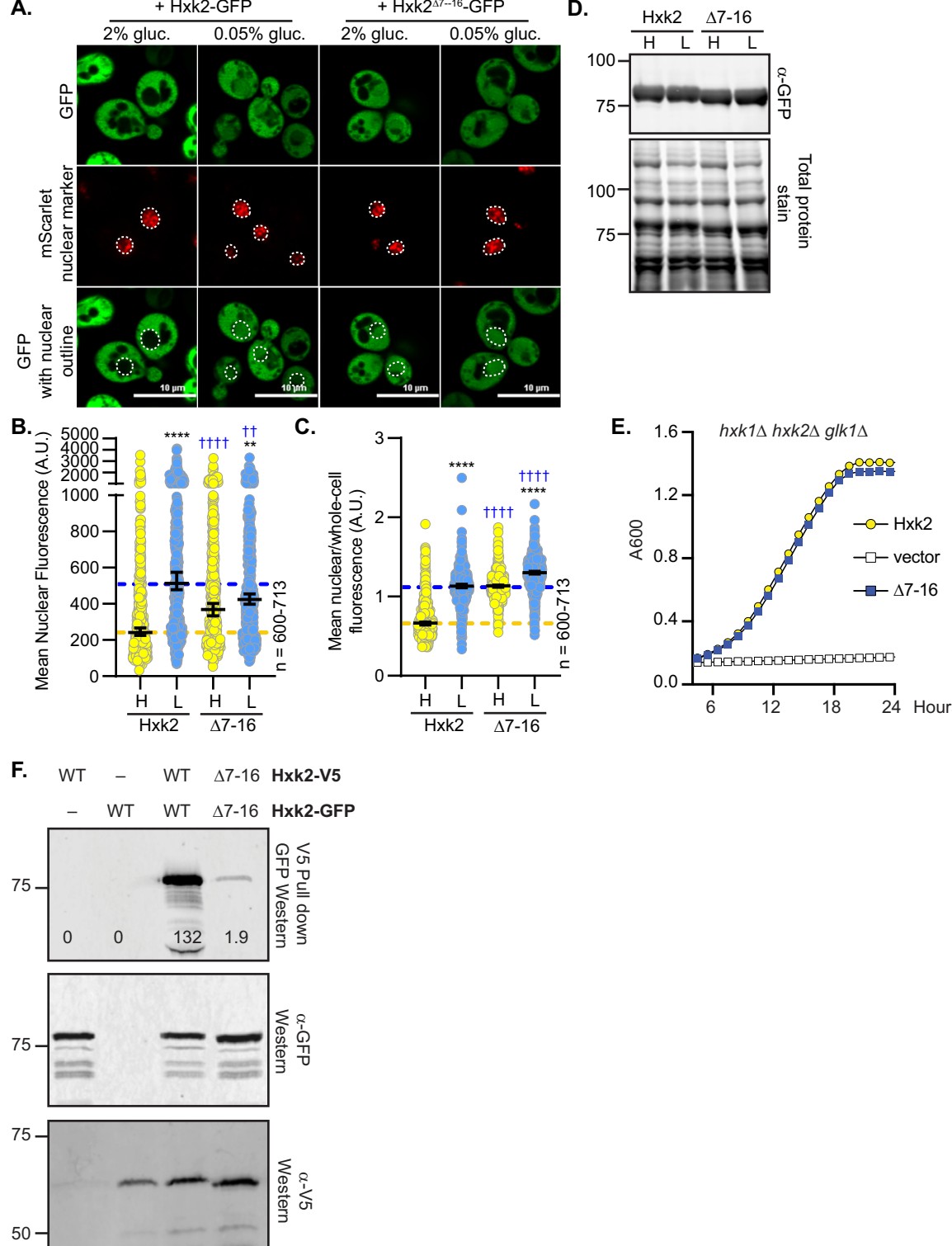

**Fig 5. Deleting the N-terminal amino acids 7–16 results in a pool of constitutively nuclear localized Hxk2, prevents Hxk2 dimerization but maintains catalytic function.** (A) Confocal microscopy of GFP-tagged Hxk2 or Hxk2$^{\Delta7-16}$ expressed from a CEN plasmid under the control of the *HXK2* promoter in *hxk2Δ* cells. Co-localization with the nucleus is determined based on overlap with the Tpa1-mScarlet nuclear marker, and a dashed white line indicates the nucleus. (B-C) Automated quantification of the images shown in panel A to measure (B) mean nuclear fluorescence or (C) the ratio of the mean nuclear/whole-cell fluorescence. Horizontal black lines

show the median, and error bars indicate the 95% confidence interval. Dashed yellow and blue lines represent the median value for Hxk2-GFP in high and low glucose, respectively. Black asterisks represent statistical comparisons between low and high glucose for a specific *HXK2* allele, and blue daggers represent statistical comparisons between mutant alleles and the corresponding WT Hxk2 in the same medium condition. (D) Immunoblot analyses of Hxk2-GFP in whole-cell protein extracts made from cells grown in high glucose or shifted to low glucose for 2 hours. REVERT total protein stain of the membrane serves as a loading control. (E) Cells lacking all three hexokinase genes (*hxk1Δ hxk2Δ glk1Δ*) were transformed with plasmid vector or plasmids expressing wild-type Hxk2 or Hxk2-Δ7–16, as indicated. Cell growth (A$_{600}$) in media containing glucose as the carbon source was monitored for 24 hours. (F) To assess multimerization, we prepared extracts from yeast cells expressing the untagged Hxk2 or Hxk2 tagged with either V5 or GFP. Hxk2 proteins contained either the wild-type (WT) N-terminus or the Δ7–16 deletion. Protein expression was monitored by western blotting (bottom two panels). The association of the tagged proteins was assessed by co-immunoprecipitation using anti-V5 beads followed by western blotting with anti-GFP (top panel).

glucose-replete and glucose-starved conditions (Fig 6C–6E). In glucose-grown cell, it had elevated nuclear fluorescence and a nuclear to whole-cell fluorescence ratio that was indistinguishable from wild-type Hxk2 under glucose-starvation conditions. Further, there was little change in the mean nuclear to whole-cell fluorescence ratio of Hxk2$^{K13A}$ between glucose-grown or -starved cells (Fig 6E), demonstrating that this mutant bypasses glucose-inhibition of nuclear localization. A mutation of K13 to arginine, which would disrupt sumoylation and likely disrupt methylation (arginine's can be methylated but use differing enzymes than lysine methylation) but could leave Hxk2 dimerization intact, similarly prevented nuclear exclusion of Hxk2 in high glucose conditions (S7D–S7E Fig). These data suggest that loss of sumoylation and/or methylation at K13 may be an important determinant for glucose restriction of Hxk2's nuclear translocation. The analogous Hxk1$^{K13A}$ mutant did nothing to change the distribution of Hxk1, which remained cytosolic in all conditions (S8A–S8C Fig).

Hxk2$^{K13A}$ encodes a stable and functional protein, has catalytic activity, and permits robust growth on glucose when expressed as the only hexokinase in cells (Fig 6F–6G). There was no increase in the free-GFP breakdown product for K13A relative to the WT control, so the elevated nuclear fluorescence is not due to increased free-GFP (S7G Fig). The K13A mutation disrupted multimer formation as effectively as Hxk2$^{S15D}$ (Figs 3G and 6H); V5-tagged Hxk2$^{K13A}$ from yeast extracts could not copurify GFP-tagged Hxk2$^{K13A}$ (Fig 6H).

It should be noted that the K13A mutation and dimer disruption allow access to the S15 phosphorylation site, which others have suggested might control nuclear translocation [11]. To determine if S15 phosphorylation influenced the misregulated localization of the K13A mutant, we combined K13A with S15A or S15D, which prevent phosphorylation or mimic phosphorylation, respectively. Neither of these new double mutants–Hxk2$^{K13A,S15A}$ nor Hxk2$^{K13A,S15D}$ –showed any difference in nuclear localization compared to Hxk2$^{K13A}$ (Fig 6C–6E). We propose that S15 phosphorylation is dispensable for nuclear translocation and/or retention of Hxk2$^{K13A}$.

We mutated D106, the residue that pairs with K13 to form a salt bridge at the Hxk2 dimer interface (Fig 3G). Interestingly, the Hxk2$^{D106A}$ mutant, which would presumably break the interaction with K13 and allow it to be accessible for modification and/or binding, maintained normal Hxk2 cytosol-nuclear partitioning in both glucose-replete and -starvation conditions (Fig 7A–7C). However, as anticipated, the D106A mutant could not form multimers *in vivo* (Fig 7D), supporting the idea that K13 should be accessible in this mutant. These data demonstrate that generating the Hxk2 monomer is not sufficient to drive nuclear localization of Hxk2. In addition, access to K13, as would be expected to occur in any mutant that breaks the dimer, is not enough to stimulate nuclear localization, supporting a model where post-translational modification or some aspect specific to the lysine at this position is required for glucose regulation of Hxk2's nuclear propensity.

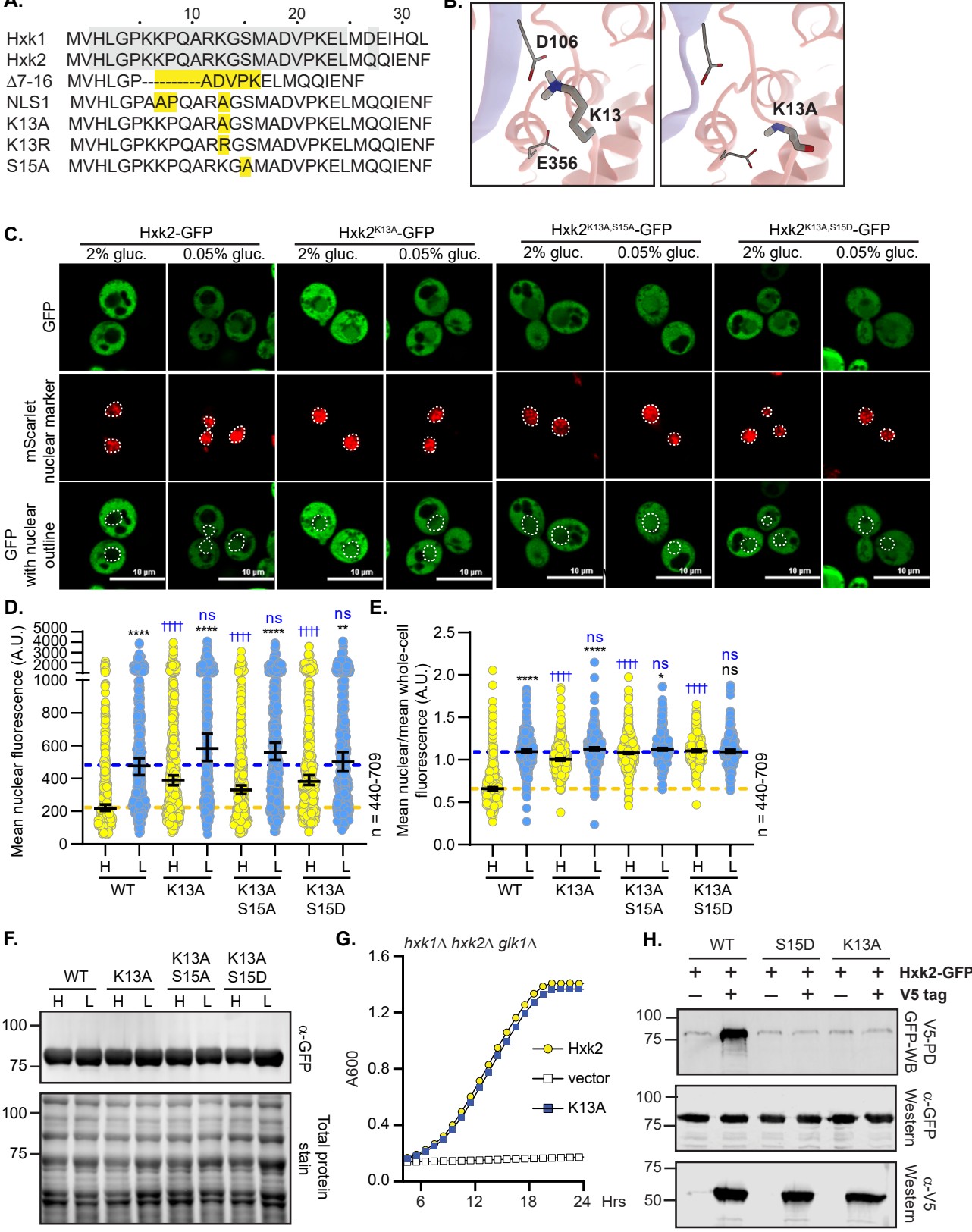

**Fig 6. Mutation of K13 to alanine in the Hxk2 N-terminal tail results in a pool of nuclear localized Hxk2 in both high and low glucose and prevents Hxk2 dimerization but maintains catalytic function.** (A) Sequence alignment of the Hxk2 N-terminal tail with Hxk1, highlighting key mutated residues. The first 30 residues of Hxk1 and Hxk2 are shown with identical residues shaded. Select mutations in this region are shown in yellow. (B) Modeled K13 interactions. The two Hxk2 monomers are shown as pink and blue ribbons. In the first panel, a close-up view of K13 interactions with E356 (same monomer) and D106 (opposing monomer). In the second panel, an alanine substitution at K13 cannot form the same electrostatic interactions. (C) Confocal microscopy of GFP-tagged Hxk2 or the mutant alleles expressed from a CEN plasmid under the control of the *HXK2* promoter in *hxk2Δ* cells. Co-localization with the nucleus is determined based on overlap with the Tpa1-mScarlet nuclear marker, and a dashed white line indicates the nucleus. (D-E) Automated quantification of imaging shown in (C) to measure (D) mean nuclear fluorescence or (E) the ratio of the mean nuclear/whole-cell fluorescence. Horizontal black lines show the median, and error bars indicate the 95% confidence interval. Dashed yellow and blue lines represent the median value for Hxk2-GFP in high and low glucose, respectively. Black asterisks represent statistical comparisons between low and high glucose for a specific *HXK2* allele, and blue daggers represent statistical comparisons between mutant alleles and the corresponding WT Hxk2 in the same medium condition. (F) Immunoblot analyses of Hxk2-GFP in whole-cell protein extracts made from cells grown in high glucose or shifted to low glucose for 2 hours. (G) Cells lacking all three hexokinase genes (*hxk1Δ hxk2Δ glk1Δ*) were transformed with plasmid vector or plasmids expressing wild-type Hxk2 or Hxk2$^{K13A}$, as indicated. Cell growth ($A_{600}$) in media containing glucose as the carbon source was monitored for 24 hours. (H) Extracts were prepared from yeast cells expressing Hxk2-GFP and Hxk2 with or without the V5 as indicated. Hxk2 proteins contained either the wild-type (WT) sequence or the S15D or K13A mutations. Protein expression was monitored by western blotting (bottom two panels). The association of the tagged proteins was assessed by co-immunoprecipitation using anti-V5 beads followed by western blotting with anti-GFP (top panel). Quantitation of the signal in the top panel is shown.

## Tda1, but not Snf1 or Mig1, is required for Hxk2 nuclear accumulation

Many studies have identified Hxk2 S15 phosphorylation [30,55–60]. This site is conserved perfectly in Hxk1, where it is also phosphorylated [55–61]. Multiple kinases are linked to Hxk2 S15 phosphorylation, including PKA, Snf1, and Tda1 [11,29,30]. The Snf1 kinase, its substrate Mig1, and Reg1 (an activator of the PP1 protein phosphatase Glc7 that controls Snf1 activity) are thought to interact with Hxk2 to facilitate its nuclear translocation, where they form a large complex that controls Hxk2's alleged moonlighting function as a transcriptional regulator [7–11].

We examined the impact of Snf1 and Mig1 on Hxk2 nuclear propensity. In the absence of Snf1 or Mig1, there was little change in Hxk2 nuclear localization in glucose-replete or -starvation conditions (Fig 8A–8C). Upon glucose starvation, *snf1Δ* and *mig1Δ* cells had slightly reduced or elevated mean nuclear fluorescence, respectively, compared to WT cells, but these changes were either not significant or just past the significance threshold (Fig 8A–8B). When the nuclear to whole-cell fluorescence ratio was considered, *mig1Δ* cells were not different than WT under any condition, while *snf1Δ* cells had increased Hxk2 nuclear balance in high glucose conditions (Fig 8C). These results counter earlier findings, which suggested Mig1 is required for Hxk2 nuclear translocation [8].

We attempted to recapitulate the reported co-purification of Hxk2 with Mig1. However, we could not capture any HA-tagged Mig1 above the weak background signal of bead binding using a Hxk2-V5 pulldown (S9A Fig). Although negative data, they are consistent with our other observations suggesting *mig1Δ* does not impact Hxk2 nuclear translocation.

The Tda1 kinase is important for phosphorylating Hxk2 S15, thereby preventing Hxk2 dimerization [23,62,63]. Consistent with a role for Tda1 in Hxk2 regulation, glucose-starvation-induced Hxk2 nuclear accumulation was severely dampened in *tda1Δ* cells (Fig 8D–8F). Recent biochemical fractionation studies found that Tda1 becomes nuclear in glucose-starved cells [64]. In our experiments, the mNeonGreen-tagged Tda1 signal was low in glucose-replete conditions, with a 3-fold increase in the mean nuclear and whole-cell fluorescence upon glucose starvation (Fig 9A–9C). However, the nuclear-to-whole-cell fluorescence ratio was only modestly different in glucose-grown versus -starved cells demonstrating that the nuclear-cytosolic balance of Tda1 was maintained in these two conditions (Fig 9D). Immunoblot analyses showed a significant increase in Tda1-mNG abundance in response to glucose starvation (Fig 9E).

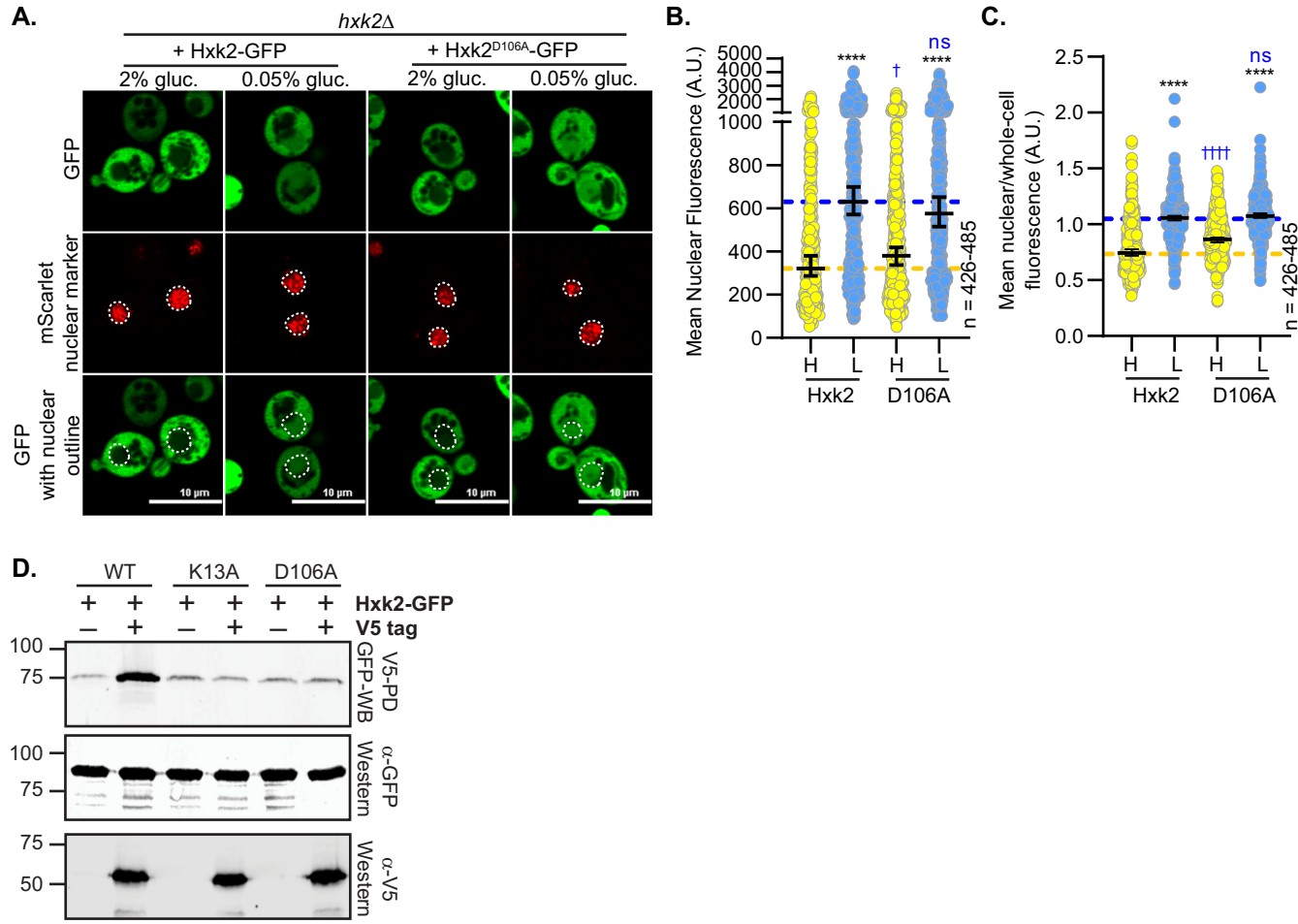

**Fig 7. Mutation of D106, which interacts with K13, does not alter Hxk2 nuclear localization but does prevent Hxk2 dimerization.** (A) Confocal microscopy of GFP-tagged Hxk2 or the mutant alleles expressed from a CEN plasmid under the control of the *HXK2* promoter in *hxk2Δ* cells. Co-localization with the nucleus is determined based on overlap with the Tpa1-mScarlet nuclear marker, and a dashed white line indicates the nucleus. (B-C) Automated quantification of imaging shown in (A) to measure (B) mean nuclear fluorescence or (C) the ratio of the mean nuclear/whole-cell fluorescence. Horizontal black lines show the median, and error bars indicate the 95% confidence interval. Dashed yellow and blue lines represent the median value for Hxk2-GFP in high and low glucose, respectively. Black asterisks represent statistical comparisons between low and high glucose for a specific *HXK2* allele, and blue daggers represent statistical comparisons between mutant alleles and the corresponding WT Hxk2 in the same medium condition. (D) Extracts were prepared from yeast cells expressing Hxk2-GFP and Hxk2 with or without the V5 as indicated. Hxk2 proteins contained either the wild-type (WT) sequence or the K13A or D106A mutations. Protein expression was monitored by western blotting (bottom two panels). The association of the tagged proteins was assessed by co-immunoprecipitation using anti-V5 beads followed by western blotting with anti-GFP (top panel). Quantitation of the signal in the top panel is shown.

Increased Tda1 levels were not driven by elevated transcription, as RNAseq analyses showed no change in *TDA1* transcript abundance for cells grown in 2% or 0.05% glucose (Fig 9F), unlike the glucose-responsive transcripts of hexose transporters 3 and 6 (*HXT3* and *HXT6*, respectively) shown as controls. As expected [65,66], *HXT3* transcripts were reduced upon glucose starvation, and *HXT6* transcripts increased (Fig 9F). The increased Tda1 could be due to altered translation or a change in protein stability/regulation.

Consistent with altered regulation, immunoblots showed a sizeable shift to a slower-mobility Tda1 in low-glucose conditions (Fig 9E). This mobility change was due to phosphorylation, as incubation with phosphatase generated a single band that migrated like Tda1 in glucose-grown cells (Fig 9G). Recent studies identified several Snf1-dependent phosphorylation sites in Tda1, so perhaps the increased phosphorylation of Tda1 under glucose starvation conditions is partially regulated by Snf1 [67]. We propose that Tda1 is required for Hxk2 nuclear

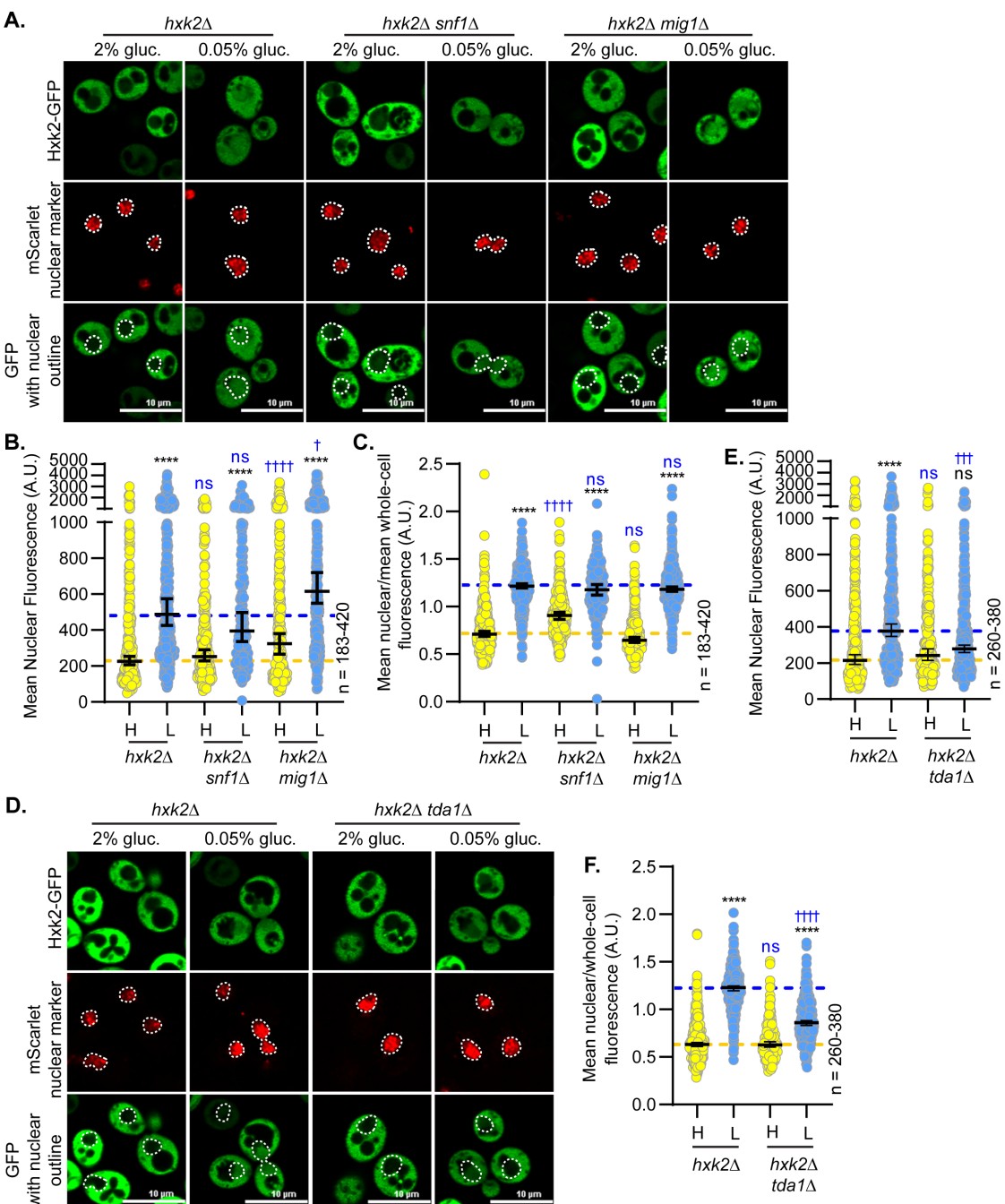

**Fig 8. Hxk2 nuclear localization is not regulated by Snf1 or Mig1 but is controlled by the Tda1 kinase.** (A and D) Confocal microscopy of GFP-tagged Hxk2 expressed from a CEN plasmid under the control of the *HXK2* promoter in cells lacking *HXK2* alone or further missing (A) *SNF1* or *MIG1* or (D) *TDA1*. Co-localization with the nucleus is determined based on overlap with the Tpa1-mScarlet nuclear marker, and a dashed white line indicates the nucleus. (B-C; E-F) Automated quantification of the images shown in panels A or D to measure (B or E, respectively) mean nuclear fluorescence or (C or F, respectively) the ratio of the mean nuclear/whole-cell fluorescence. Horizontal black lines show the median, and error bars indicate the 95% confidence interval. Dashed yellow and blue lines represent the median value for Hxk2-GFP in high and low glucose, respectively. Black asterisks represent statistical comparisons between low and high glucose for a specific *HXK2* allele, and blue daggers represent statistical comparisons between mutant alleles and the corresponding WT Hxk2 in the same medium condition.

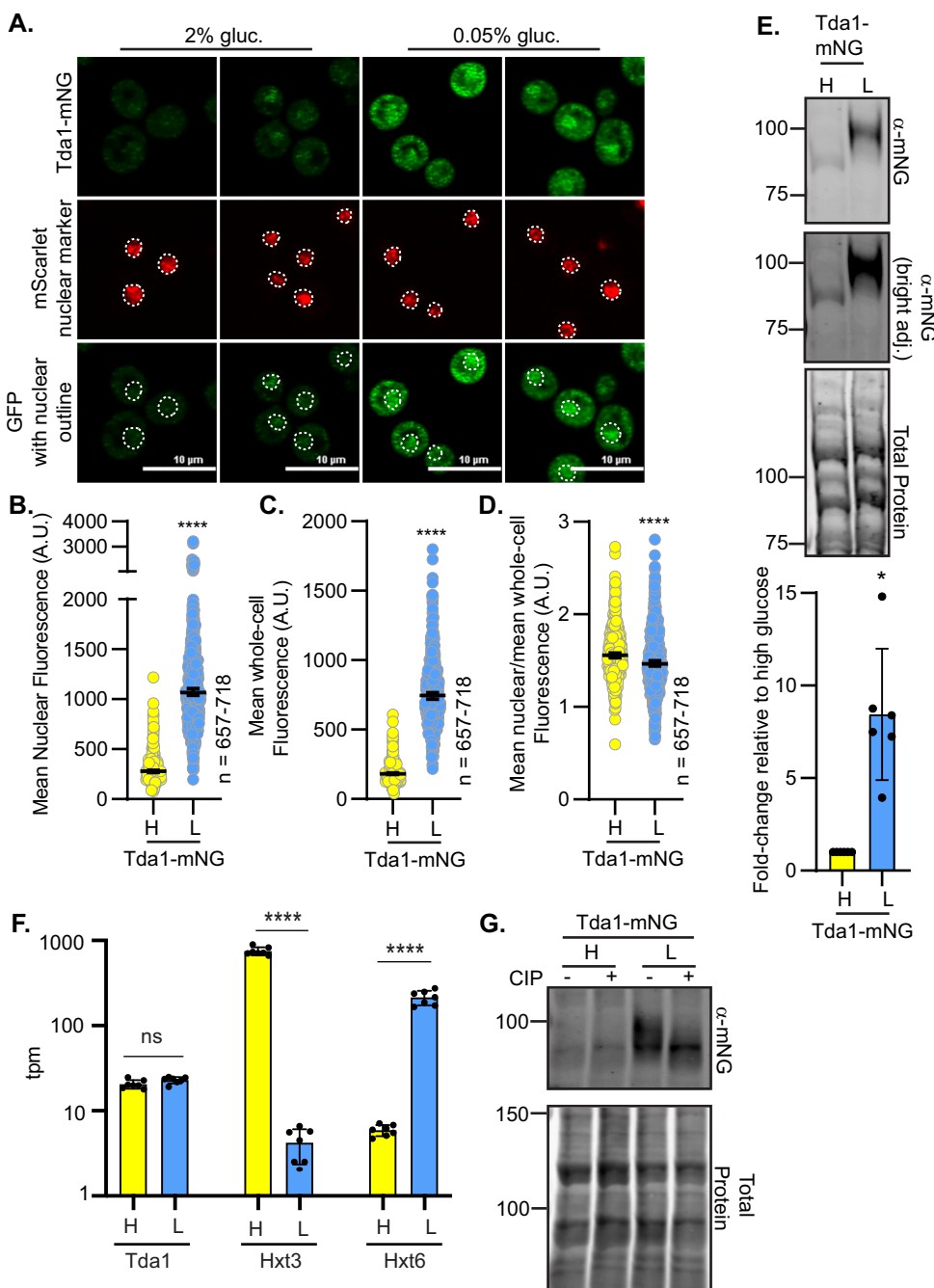

**Fig 9. Tda1 protein abundance and phosphorylation are increased in response to glucose starvation.** (A) Confocal microscopy of chromosomally integrated Tda1-mNG. Co-localization with the nucleus is determined based on overlap with the Tpa1-mScarlet nuclear marker, and a dashed white line indicates the nucleus. (B-D) Automated quantification of the images shown in panel A to measure (B) mean nuclear fluorescence, (C) mean whole-cell fluorescence, or (D) the ratio of the mean nuclear/whole-cell fluorescence. Horizontal black lines show the median, and error bars indicate the 95% confidence interval. A Student's t-test was used to assess significance (**** = p-value < 0.0001). (E) Immunoblot analyses of Tda1-mNG in whole-cell protein extracts made from cells grown in high glucose or shifted to low glucose for 2 hours. REVERT total protein stain of the membrane serves as a loading control. One representative blot from 6 biological replicates is shown. Quantification of Tda1-mNG abundance is presented as a fold-change in Tda1 levels after a 2 h shift to low glucose in comparison to glucose-grown cells (high glucose). (F) Yeast mRNA abundance (transcripts per million mapped reads; tpm) was measured by RNAseq in cells grown in high glucose (H) or two hours after shifting to low glucose, L. The mean tpm values (±SD) of multiple replicates for each condition are plotted for three genes: *TDA1*, *HXT3*, and *HXT6*. Statistical differences between the values in high and

low glucose are indicated. (G) Whole-cell extracts were made from strains expressing Tda1-mNG and grown in either 2% glucose, H, or shifted for 2 h into low glucose, L (0.05% glucose). Extracts were either treated with calf intestinal alkaline phosphatase (CIP) or incubated in the same conditions without enzyme (mock), resolved by SDS-PAGE, and immuno-blotted with anti-mNG antibody. REVERT total protein stain is shown as a loading control. Molecular weights are shown on the left side in kDa.

accumulation in glucose-starved cells. Tda1 is a glucose-regulated kinase, and glucose starvation increases its protein abundance and phosphorylation. Elevated Tda1 levels likely reflect post-transcriptional regulation since Tda1 transcript abundance is unchanged in glucose-starved cells.

We next examined the impact of Hxk2 mutations on Tda1-dependent Hxk2 nuclear translocation. As before, Hxk2 accumulation in the nucleus was blunted in glucose-starved $tda1\Delta$ cells (Figs 8D–8F and 10A–10C). However, the S15D mutation restored glucose-starvation-induced Hxk2 nuclear localization in $tda1\Delta$ cells (Fig 10A–10C). The balance of Hxk2$^{S15D}$ or Hxk2$^{S15A}$ nuclear-to-whole-cell fluorescence ratios in $tda1\Delta$ cells was similar to those of Hxk2 in wild-type cells but shifted higher in Hxk2$^{S15D}$ $tda1\Delta$ or Hxk2$^{S15D}$ $tda1\Delta$ cells in high glucose conditions. Unlike Hxk2$^{S15D}$ or Hxk2$^{S15A}$, the Hxk2$^{K13A}$ mutation bypassed both Tda1 and glucose-starvation regulation. Hxk2$^{K13A}$ was nuclear in glucose-grown $tda1\Delta$ cells, though its overall nuclear abundance was somewhat reduced compared to glucose starvation (Fig 10B), and Hxk2$^{K13A}$ retained a larger nuclear pool than WT Hxk2 in both glucose-grown and -starved cells (Fig 10A–10C). There were no changes in the free-GFP breakdown product at 2 h post low glucose shift in the $tda1\Delta$ cells containing K13A (S7G Fig).

Although the Hxk2$^{S15D}$ mutation restored nuclear localization in glucose-starved $tda1\Delta$ cells, this mutation had a different effect in $snf1\Delta$ cells (S10A–S10C Fig). We observed a higher nuclear-to-whole-cell ratio in glucose-grown $snf1\Delta$ cells expressing Hxk2$^{S15D}$ than wild-type Hxk2 (S10C Fig). Interestingly, this increase in nuclear-to-whole-cell fluorescence was not driven by higher mean nuclear fluorescence, which is the same for Hxk2$^{S15D}$ and Hxk2 in $snf1\Delta$ or wild-type cells (S10B Fig). These findings suggest the S15D mutation bypasses Tda1 regulation, restoring glucose-regulated Hxk2 nuclear accumulation. However, unlike the Hxk2$^{K13A}$ mutant, the Hxk2$^{S15D}$ mutation does not alter Hxk2 localization in glucose-grown cells.

Since Hxk2$^{S15D}$, Hxk2$^{D106A}$, and Hxk2$^{K13A}$ all disrupt Hxk2 dimerization *in vivo*, yet only Hxk2$^{K13A}$ prevents glucose-regulated nuclear exclusion, Hxk2 monomer-dimer balance cannot the only thing controlling Hxk2 nuclear partitioning. Hxk2$^{S15D}$, which is monomeric, does not change Hxk2 nuclear partitioning in glucose-grown cells. Instead, it is only important for glucose-starvation-induced Hxk2 nuclear localization in the $tda1\Delta$. In contrast, Hxk2$^{K13A}$, which is also monomeric, bypasses the glucose regulation of Hxk2 nuclear partitioning and this is not altered by modification at S15.

## Role of Hxk2 in regulating glucose-repression of gene expression

Earlier studies proposed that Hxk2 is a subunit of a DNA-bound repressor complex that regulates glucose repression of gene expression [7,15]. We performed RNA-seq analyses of the yeast transcriptome in wild type and $hxk2\Delta$ cells grown in high glucose or shifted to glucose-limiting conditions (0.05% glucose) for two hours. RNA samples were prepared in triplicate, and the log2+1 ratios (low glucose/high glucose) for the mean abundance of each mRNA were plotted (Fig 11A–11D). As reported previously [4,68], the yeast transcriptome undergoes large-scale changes in transcript levels in response to glucose limitation, with >15% of the transcripts showing a 4-fold or higher change in abundance (Fig 11A). Notably, large changes

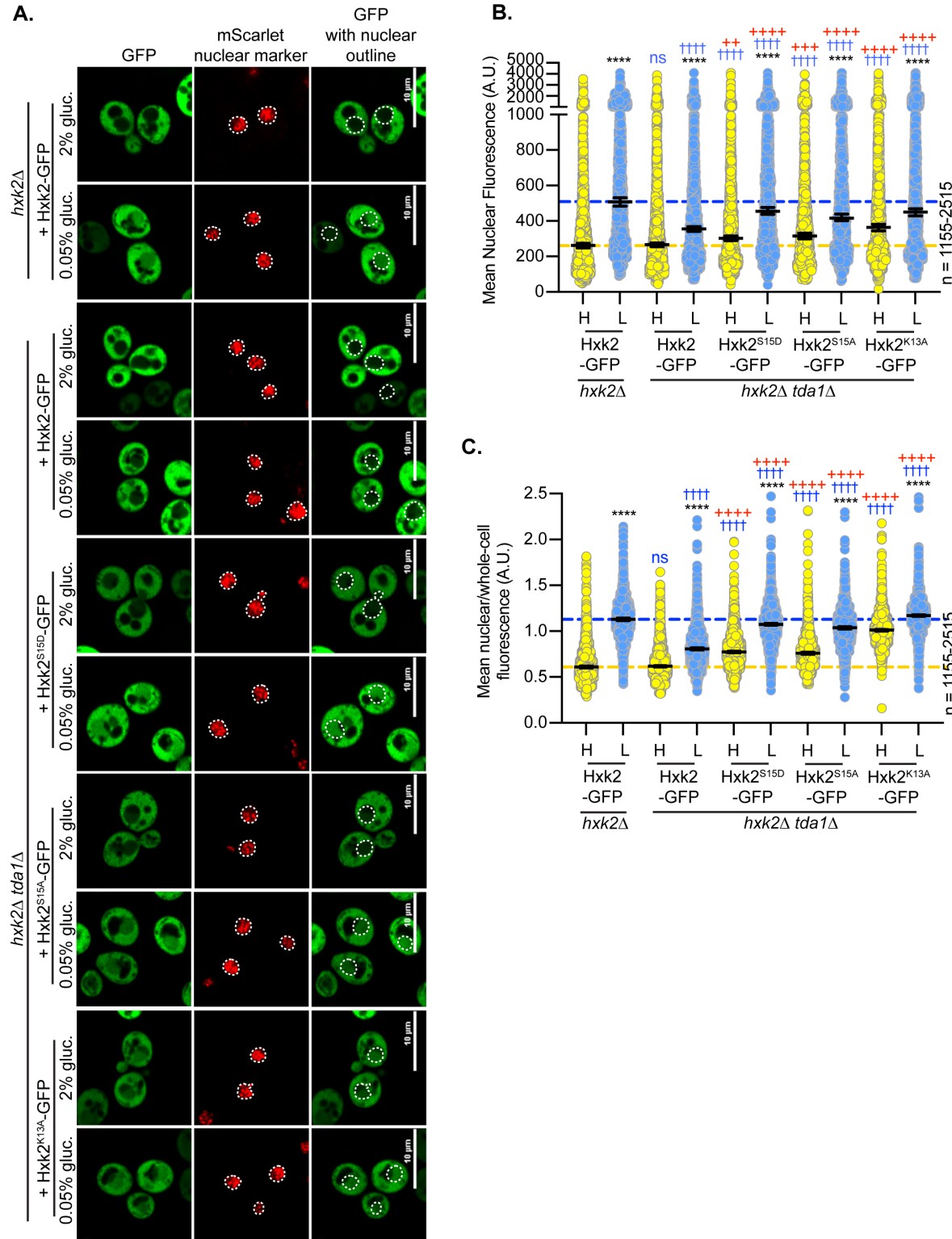

**Fig 10. Mutating Hxk2 S15 to aspartic acid rescues the impaired nuclear localization caused by a *tda1Δ*. (A)** Confocal microscopy of GFP-tagged Hxk2 and S15D mutant expressed from a CEN plasmid under the control of the *HXK2* promoter in cells lacking *HXK2* alone or with the additional deletion of *TDA1*. Co-localization with the nucleus is determined based on overlap with the Tpa1-mScarlet nuclear marker, and a dashed white line indicates the nucleus. (B-C) Automated quantification of the images shown in panel A to measure (B) mean nuclear fluorescence or (C) the ratio of the mean nuclear/whole-cell fluorescence. Horizontal black lines show the median, and error

bars indicate the 95% confidence interval. Dashed yellow and blue lines represent the median value for Hxk2-GFP in high and low glucose, respectively. Black asterisks represent statistical comparisons between low and high glucose for a specific *HXK2* allele. Blue daggers represent statistical comparisons between mutant alleles and the corresponding WT Hxk2 in the same medium condition. Red crosses represent statistical comparisons between WT Hxk2 and mutants in the *hxk2Δ tda1Δ* background in the same media condition.

(both increases and decreases) in the abundance of the hexose transporter (*HXT*) mRNAs were observed, as well as a decrease in the abundance of ribosomal protein mRNAs. A comparable pattern, scale, and scope of mRNA abundance changes were observed in the RNA samples generated from *hxk2Δ* cells (Fig 11B), demonstrating that the Hxk2 protein is not required for the global transcriptional response to changes in glucose abundance. Replotting these values to show the ratio of mRNA abundance (*hxk2Δ*/wild type) in high glucose (Fig 11C) and low glucose (Fig 11D) further demonstrated that *HXK2* deletion impacts very few mRNAs. Previous studies have suggested that Hxk2 is involved in glucose repression based on changes in a small number of genes (*SUC2, HXT1, HXK1, CYC1, etc.*) [15,16,65]. We also see similar expression changes in our RNAseq data with *hxk2Δ* to those previously reported (S11A Fig). However, we find that glucose repression is largely unaffected by *hxk2Δ*, suggesting that the small number of genes whose transcription changes in *hxk2Δ* cells are not representative of a global loss of glucose repression. A violin plot of the log2+1 ratios from this experiment demonstrated the large and similar response to glucose limitation in both wild-type and *hxk2Δ* cells (Fig 11E). By comparison, the difference between wild-type and *hxk2Δ* cells under these conditions was modest, with nearly no 2-fold or greater changes in transcript abundances (Fig 11E). Next, we plotted the mean mRNA abundance of the top 20 glucose-repressed genes in high and low glucose in wild type and *hxk2Δ* cells (Fig 11F). Glucose repression and derepression of these mRNAs were not affected by deletion of *HXK2*, even for the genes whose promoters are bound by the Mig1 protein [69]. When we analyzed Mig1-regulated genes [69] with a 2-fold or greater change in transcript abundance in response to low glucose, we found no evidence that any of these were altered in *hxk2Δ* cells (S11B Fig). These data support a model where Hxk2 is not a transcriptional regulator, and does not regulate glucose repression.

## Discussion

Glucokinases and hexokinases are central metabolic regulators that convert glucose to G6P, the first step in glycolysis. In addition to cytosolic functions in glycolysis, each enzyme of this pathway can accumulate in the nucleus. For mammalian and plant hexokinases, nuclear accumulation typically occurs in glucose starvation or other stress conditions [24,25,70–72]. What is the nuclear function of glycolytic enzymes? Perhaps they act together in nuclear glycolysis to regulate a nuclear pool of ATP or NADH [63,73]. Alternatively, could they have a nuclear role in regulating gene expression or other facets of nuclear biology? The nuclear function of many glycolytic enzymes remains unclear [73]. For a handful of examples, there is evidence for diverse roles that involve altering transcription factor function, associating with RNA polymerase III, interacting with DNA, regulating nuclear ubiquitin ligases, stimulating DNA polymerase, and protecting telomeres [74–78]. In some instances, the metabolic products generated by glycolytic enzymes may be the active nuclear component rather than the proteins [79]. Much remains to be discovered about the "moonlighting" nuclear activities of glycolytic enzymes.

In studies of hexokinase localization, *S. cerevisiae* and *C. albicans* hexokinase 2 have been outliers. Their nuclear accumulation reportedly occurs in glucose-replete rather than starvation conditions [8,11,13,14,34]. However, the methodologies used in early imaging studies

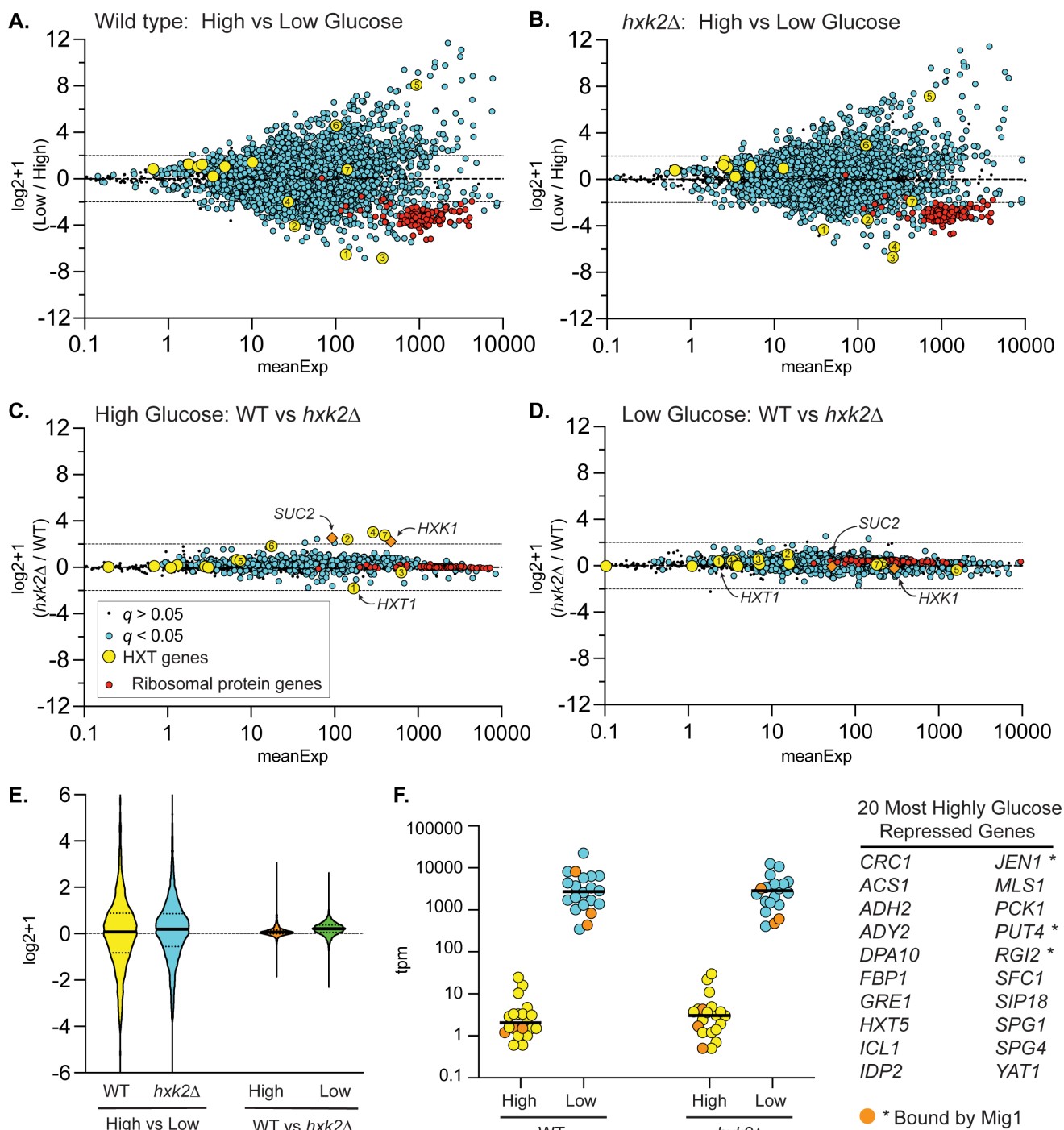

**Fig 11. Effect of Hxk2 on the transcriptional response to glucose limitation.** Total RNA was collected from 3 independent cultures of wild-type cells or *hxk2Δ* cells grown to mid-log phase in high glucose or two hours after shifting to low glucose. mRNA abundance (tpm) for all genes was determined by RNAseq and is plotted as the mean expression level (x-axis) versus the log2+1 value of the ratio of the mean value in low glucose divided by the mean value in high glucose (A and B) in wild type (A) or *hxk2Δ* cells (B). The same data are replotted, showing the log2+1 ratio of *hxk2* and wild-type cells in high (C) and low glucose (D) conditions. In each plot, statistical significance was defined as a false discovery rate (q) less than 0.05. Log2+1 values deemed significant were plotted as blue circles, while those failing to meet this threshold were plotted as smaller black dots (see key). *HXT* gene mRNAs were plotted as yellow circles with the number referring to the specific *HXT* gene. Ribosomal mRNA genes were plotted as red circles. The known Hxk2-regulated genes, *SUC2* and *HXK1*, were plotted as orange diamonds. (E) Violin plot of these data showing the relative magnitude of glucose limitation (high vs. low) and *hxk2* deletion (WT vs. *hxk2Δ*). (F) The top 20 glucose-repressed genes (listed at right) are plotted based on their TPMs from both high glucose (yellow) and low glucose (blue) conditions. Three of these genes (indicated by * in the table at right and shown as orange-filled circles on the plot) are reported to have Mig1 bound in their promoters based on ChIP-Seq.

involved cells pre-incubated in a way that would induce glucose starvation. This confounds the interpretation of these studies, especially if the incubation time in the glucose-starvation medium was not regimented [8,11,13,14,34].

## Contrasts between our findings and the existing model for Hxk2 nuclear regulation

Here we report high-resolution, live-cell, confocal microscopy of fluorescently tagged Hxk2 performed in high (2% glucose) and low (0.05% glucose) glucose medium. Our findings support a new model for Hxk2 nuclear translocation and suggest that in yeast, Hxk2 accumulates in the nucleus under glucose-starvation conditions, not in glucose-replete conditions. This is consistent with a conserved model for glucose-regulated hexokinase nuclear localization spanning the ~1 billion years of evolution that separate yeasts and humans [80,81].

In contrast to earlier models, we find that: (1) Mig1 is not required for Hxk2 nuclear translocation; (2) the previously reported NLS/Mig1 binding site (amino acids 7–16) is not required for Hxk2 nuclear translocation but instead maintains a glucose-regulated, nuclear-excluded pool of Hxk2; (3) Snf1 is not required for glucose-regulated nuclear localization of Hxk2; (4) phosphorylation of S15, though a key regulator of the Hxk2 monomer-dimer balance, is not required for glucose-regulated Hxk2 nuclear accumulation; and (5) except for modest changes in a handful of transcripts, Hxk2 is not required to maintain glucose-repression of transcription.

The past model for Hxk2 nuclear shutting was based on studies from a single lab that others have not corroborated. For example, large-scale protein-protein association studies using TAP-purification and mass spectrometry [37,82] or global two-hybrid screens [83,84] have not detected Hxk2 association with the components of the proposed Hxk2 nuclear complex. Targeted studies from other labs have not provided secondary confirmation of this Hxk2-containing complex or the glucose-stimulated nuclear translocation of Hxk2. Comprehensive ChIP-Seq data fail to identify this complex at the *SUC2* promoter [69], a gene regulated by glucose repression whose transcription is reportedly controlled by Hxk2 [7,15,85,86]. The role of S15 phosphorylation in regulating Hxk2 nuclear translocation has been openly questioned [23]. Our data, together with these observations, suggest that the previous model of Hxk2 nuclear accumulation in yeast is not correct.

## A new model for regulation of Hxk2 nuclear translocation

Based on our studies, we propose a new model for glucose-regulated Hxk2 nuclear accumulation (Fig 12). In a glucose-replete environment (Fig 12A), the Glc7-Reg1 phosphatase is active, maintaining inactive Snf1 [87,88]. The Tda1 kinase, though transcribed, is in low abundance in cells grown in 2% glucose, suggesting it is either not translated or is an unstable protein. Under these conditions, Hxk2 S15 is not phosphorylated, and Hxk2 exists in cells as a balance between monomer and dimer species [31,32].

We find that Hxk2 shifts to a monomer when glucose binds, confirming earlier studies that demonstrate a dramatic decrease in the association constant of Hxk2 dimers when glucose is present [32]. Molecular dynamics simulations provide insight into why glucose might favor Hxk2 monomer formation (S1 Text). In our simulations, bound glucose impacts multiple electrostatic interactions between the opposite monomer's N-terminal tail and the catalytic pocket, which may promote Hxk2 dimer dissociation. Our experimental evidence confirms that the N-terminal tail is critical for dimerization. Deleting amino acids 7–16 or mutating K13 to alanine both give rise to monomeric Hxk2. Disrupting the K13-D106 salt bridge at the dimer interface by alanine substitution at D106 also breaks the dimer but does not stimulate the nuclear accumulation of Hxk2 in glucose-replete conditions.

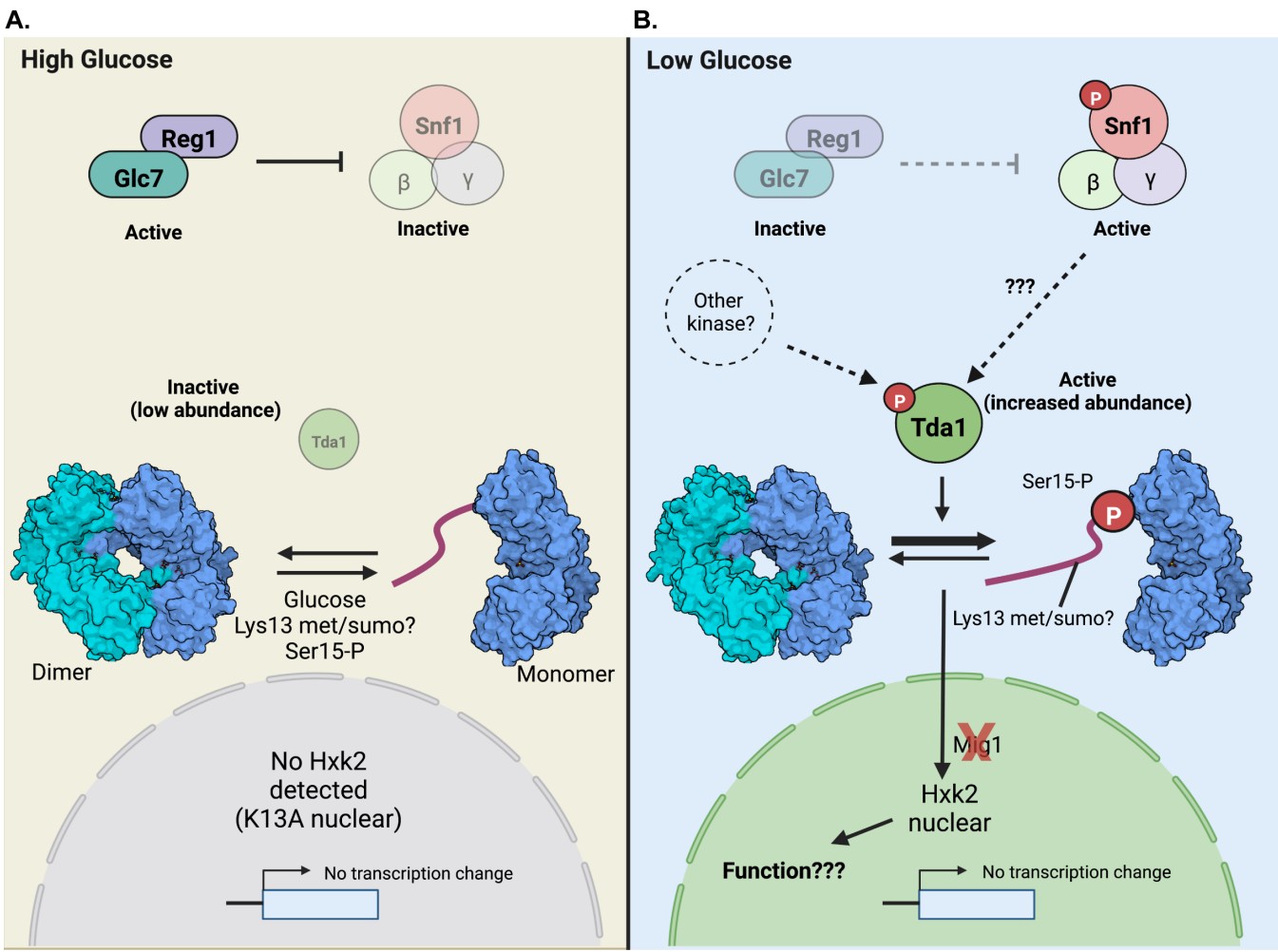

**Fig 12. Model for glucose-regulation of Hxk2 nuclear accumulation and dimerization.** (A) A schematic of key regulatory factors and their function when cells are grown in glucose replete (referred to as high glucose) conditions. For cells grown in glucose, the Reg1-Glc7 protein phosphatase complex is active, and this dephosphorylates Snf1 to keep it inactive. The impact of this regulation on Tda1 is currently unclear. The Tda1 kinase is present in cells at very low levels and is an inactive kinase. Glucose binding to the Hxk2 dimer (the two Hxk2 monomers are shown in light and dark blue, respectively) stimulates monomer formation, as does mutation of the enzyme at S15, K13 and D106. Under these conditions, no nuclear Hxk2 is detected unless the K13A mutant is present, which somehow inactivates the glucose-induced block to Hxk2 nuclear accumulation. From the RNA-seq analyses, Hxk2 does not regulate glucose-repressed gene expression. (B) A schematic of key regulatory factors and their function when cells are glucose starved (referred to as low glucose). Under these conditions, Reg1-Glc7 is inactivated. However, we find that Reg1 is needed for Hxk2 nuclear translocation in this condition, but the mechanism underlying this requirement is unclear. Snf1 kinase is phosphorylated and activated. It is unclear if Snf1 is responsible for phosphorylating Tda1 to activate it kinase. While loss of Snf1 did not alter Hxk2 nuclear accumulation in low glucose, prior work demonstrates that Snf1 controls Tda1 phosphorylation in alternative carbon sources and that loss of Snf1 reduces the amount of Hxk2 monomer in cells. If Snf1 is involved in this pathway, its function may be redundant with other kinases that operate in this pathway. In glucose starvation conditions, the abundance of Tda1 increases dramatically, and Tda1 is phosphorylated and activated. Phosphorylation of Hxk2 by Tda1 at S15 regulates the Hxk2 dimer to monomer transition. Tda1 is required for Hxk2 nuclear translocation, but Mig1 and Snf1 are not needed. Hxk2 phosphorylation at S15 reduces the capability of Hxk2 to dimerize by more than 1000-fold [31]. This stabilization of the monomeric species could allow for additional modifications at the Hxk2 N-terminus. Notably, K13 is reportedly dimethylated or sumoylated [52,53]. However, the K13R mutant, which would block sumoylation at this site, also promotes nuclear localization of Hxk2 in high glucose conditions. Thus, sumoylation cannot be required for the nuclear translocation but could be important for preventing Hxk2 nuclear localization in high glucose. Methylation can occur on either K or R residues, thus the K13R mutation does not necessarily prevent this modification. Once in the nucleus, Hxk2 does not significantly impact transcription regulation. Based on our RNA-seq analyses, there is little difference between the expression profiles of WT vs. *hxk2Δ* cells glucose starvation conditions. The function of nuclear Hxk2 remains to be determined.

Hxk2 monomer-dimer balance is likely an important facet of Hxk2 regulation, but it is not responsible for Hxk2 nuclear translocation. Mutations that induce monomer formation do not all stimulate Hxk2 nuclear accumulation. For example, Hxk2$^{S15D}$ and Hxk2$^{D106A}$ both fail

to dimerize but retain glucose-regulated nuclear translocation. On the other hand, Hxk2$^{K13A}$ and Hxk2$^{\Delta 7-16}$, which also fail to dimerize, lose glucose-regulated Hxk2 nuclear exclusion. The K13R mutation gives rise to a nuclear pool of Hxk2 in high glucose conditions. These experiments identify the Hxk2 N-terminal region as key in regulating Hxk2 nuclear localization, but via a completely different mechanism than that proposed in earlier studies [8,12]. In the absence of Hxk2, glucose repression of transcription is intact; only a few transcripts show minimal changes compared to wild-type cells (Fig 12).

In response to glucose starvation (Fig 12B), the Glc7-Reg1 phosphatase is inactive, and the Snf1 kinase is activated by phosphorylation [87,88]. However, the loss of Snf1 or its substrate Mig1 does not alter the formation of the nuclear Hxk2 pool. Similarly, the Tda1 kinase is required for starvation-induced Hxk2 nuclear accumulation. When glucose is limiting, Tda1 protein levels rise, not because of altered transcription but due to increased protein stability or elevated translation. Tda1 is phosphorylated in these conditions, which activates the kinase towards histone substrates [64]. In cells starved for glucose, Tda1 phosphorylates Hxk2 at S15 and this modification disrupts the dimer in favor of monomeric Hxk2 [29,63]. The primary role of Tda1 in glucose-starved Hxk2 regulation may be to promote Hxk2 monomer formation, as both the S15A and S15D mutants, which each disrupt the dimer, were able to restore nuclear partitioning in *tda1Δ* cells.

A second layer of glucose-induced regulation must exist for Hxk2 because Hxk2$^{S15D}$ and Hxk2$^{S15A}$, unlike Hxk2$^{K13A}$, remain nuclear excluded in glucose-grown cells. In contrast, deleting the N-terminal amino acids 7–16 or changing K13 to alanine bypasses this glucose regulation. Nuclear Hxk2 accumulation in glucose-starvation conditions is not needed for transcriptional changes, as there were little to no differences in gene expression between wild-type and *hxk2Δ* cells. Given that Hxk2 does not substantially impact global gene expression, future studies must determine its nuclear function in *S. cerevisiae*.

## Comparison of *Sc*Hxk1 and *Sc*Hxk2

The first 20 amino acids that seem important for regulating Hxk2 nuclear exclusion in high glucose are perfectly conserved in Hxk1 (Fig 6A). Hxk1 and Hxk2 are paralogs, arising from the whole-genome duplication and sharing a high degree of conservation (S4A–S4B Fig). What then drives this nuclear partitioning for Hxk2, which does not seem to occur for Hxk1? Perhaps the answer lies in posttranslational modifications of these two enzymes. For the conserved K13 residue in Hxk1 and Hxk2, there is a differential modification with K13 being ubiquitinated in Hxk1 while this residue is sumoylated or dimethylated in Hxk2 [52–54]. However, if ubiquitination at K13 was required for Hxk1 nuclear exclusion, the K13A mutation we generated should have prevented this modification and permitted nuclear accumulation. However, this was not the case. The K13A mutation in Hxk2 breaks the dimer and would also prevent sumoylation and dimethylation. Since this mutant always has a nuclear-localized pool, irrespective of glucose quantity, perhaps modification at this site is needed to maintain the nuclear exclusion of Hxk2. With the K13R mutation, glucose has lost its ability to restrict the nuclear accumulation of Hxk2. This mutation likely maintains the monomer-dimer balance but would prevent sumoylation and have a somewhat unclear impact on methylation (*i. e.*, arginine residues can be methylated, but arginine methylation uses different enzymes than lysine methylation and so it seems unlikely that this would be the case). If sumoylation or methylation at K13 in Hxk2 is needed for its glucose-induced nuclear restriction, then loss of these modifications may explain why K13A and K13R accumulate in the nucleus.

## Regulation of hexokinase nuclear localization in mammals

Most isoforms of hexo- and glucokinase in mammals and plants translocate to the nucleus. However, in many instances, the nuclear function is poorly understood [73]. Glucokinase (GCK) regulation by glucokinase regulatory protein (GKRP) is one interesting example of nuclear regulation in mammals [89]. GCK is an important regulator of glucose balance that controls glucose metabolism in the liver and pancreas and regulates insulin secretion from β-islet cells [89–91]. In response to glucose starvation, GCK in the liver binds to GKRP, which competitively inhibits glucose binding [92]. When bound to GKRP, GCK translocates into the nucleus, where it serves as a reservoir of inactive GCK that can be rapidly remobilized to the cytosol when glucose becomes available [89,92].

No yeast equivalent of GKRP has been identified, and it is unclear if Hxk2 nuclear accumulation could act as a reservoir for Hxk2 function. It seems unlikely that nuclear Hxk2 would follow this "nuclear storage model" because after prolonged starvation there is still a lot of Hxk2 the cytoplasm (S2B Fig). In addition, Hxk1 and Glk1 are both expressed in response to glucose starvation and so they can phosphorylate glucose in these conditions, negating the need for a return of Hxk2 to the cytosol.

Human HKII, the isoform with the highest sequence homology to Hxk1 and Hxk2, localizes to the mitochondria in glucose-replete conditions but then translocates to the nucleus in some cancer cells [25,70,72,93]. In one case, this HKII nuclear translocation is associated with the apoptosis-inducing factor (AIF) and phosphorylated p53, a tumor suppressor [72,94]. HKII moves with these factors from the mitochondria to the nucleus to trigger apoptosis. In yeast, Aif1 is the yeast ortholog of mammalian AIF, and loss of Hxk2 is suggested to induce apoptosis via an Aif1-dependent mechanism [95,96]. It is unclear if, like the mammalian model, Hxk2 is involved in the nuclear transition of Aif1 in yeast.

Others have found ~10% of Hxk2 activity is mitochondrially associated in yeast [97]. Mass spectroscopy aimed at identifying the mitochondrial proteome under glucose-replete conditions and in alternative carbon sources further supports the presence of a mitochondrial pool of Hxk1 and Hxk2 [98]. We did not observe a distinct mitochondrial pool of Hxk2, but the bright cytosolic Hxk2 could have masked it. Perhaps a preexisting mitochondrial Hxk2 gives rise to nuclear Hxk2 in glucose-starvation conditions, as can be the case for mammalian HKII [95,96].

## Materials and methods

### Yeast strains and growth conditions

Yeast strains are summarized in S1 Table and are typically derived from the BY4742 background of *S. cerevisiae*. Where indicated, yeast were grown in synthetic complete (SC) medium lacking the appropriate amino acid for plasmid selection [99] with ammonium sulfate as a nitrogen source, or in yeast peptone dextrose (YPD). Plasmids were introduced into yeast using lithium acetate transformation [100]. Unless indicated, yeast cells were grown at 30˚C. For experiments where cells were shifted into low glucose medium, yeast cells were first grown to mid-log phase in SC medium with 2% glucose (high glucose medium). Next, cells were washed into 0.05% glucose medium (low glucose medium), resuspended low glucose medium and incubated for 2 hours at 30˚C unless otherwise indicated.

To assess whether Hxk2 mutants were enzymatically active in cells, we performed growth assays in cells lacking chromosomal *HXK1*, *HXK2*, and *GLK1* genes. Cells lacking these hexokinases fail to grow on a glucose-containing medium because they cannot phosphorylate glucose, which is required for glycolysis [101]. These *hxk1Δ hxk2Δ glk1Δ* cells were grown in SC

media containing 2% (w/v) galactose as a carbon source. Cells were transformed with plasmids expressing the genes encoding hexokinase proteins (wild type or mutant) or an empty vector. To assay hexokinase function, we grew cells in 96-well plates with media containing galactose and then inoculated them into SC media with glucose. Absorbance at 600 nm was determined using a Synergy 2 plate reader (BioTek).

## Yeast protein extraction, CIP treatments, and immunoblot analyses

Whole-cell protein extracts were generated using the trichloroacetic acid (TCA) method [102]. An equal density of mid-log phase cells (typically ~ $5x10^7$ cells) was harvested by centrifugation, washed in water, and then resuspended in water with 0.25 M sodium hydroxide and 72 mM β-mercaptoethanol. Samples were incubated on ice, and proteins precipitated by adding 50% TCA. After further incubation on ice, proteins were collected by centrifugation, the supernatant was removed, and the proteins were solubilized in 50 μL of TCA sample buffer (40 mM Tris-Cl [pH 8.0], 0.1 mM EDTA, 8 M Urea, 5% SDS, 1% β-mercaptoethanol, and 0.01% bromophenol blue). Samples were heated to 37°C for 15 min, and the insoluble material removed by centrifugation before resolving by SDS-PAGE. For Fig 9G, 15 μL of cell lysate was further treated for 1 h at 37°C with 40 units of Quick calf intestinal alkaline phosphatase (CIP, New England Biolabs, Ipswich, MA, USA) per the manufacturer's recommendations, or mock-treated in CIP buffer without enzyme. These samples were then precipitated using 50% TCA and solubilized in SDS/Urea sample buffer as above, before analysis via SDS-PAGE. Proteins were identified by immunoblotting with a mouse anti-GFP antibody (Santa Cruz Biotechnology, Santa Cruz, CA, USA) to detect green fluorescent protein (GFP) fused to Hxk2, or mouse anti-mNeon Green (mNG) nanobody (ChromoTek, Planegg-Martinsried, Germany) to detect mNG fused to Tda1. Primary antibodies were detected using anti-mouse or anti-rabbit secondary antibodies conjugated to IRDye-800 or IRDye-680 (Li-Cor Biosciences). Revert (Li-Cor Biosciences, Lincoln, NE, USA) total protein stain was used as a loading and membrane-transfer control. Secondary antibodies or Revert staining were detected on an Odyssey CLx infrared imaging system (Li-Cor Biosciences).

## Co-immunoprecipitation assays

Hexokinase proteins were C-terminally tagged with either three copies of the V5 epitope [103] or the GFP protein and expressed from CEN plasmids under the control of their respective promoters. Cells were grown in 2% glucose to mid-log phase and harvested for protein extraction. Protein expression was monitored by immunoblotting with a 1:1000 dilution of Anti V5 Antibody (Fisher Scientific Catalog # R960-25) or a 1:1000 dilution of Anti-GFP polyclonal Antibody (Product # PA1-980A). Hxk2-3V5 and associated proteins were immunoprecipitated from glass bead extracts (250 μg total protein) using 20 μL of agarose conjugated anti-V5 antibody (Sigma; A7345). The extract and beads were incubated at 4°C overnight, then washed three times in 1 mL of hexokinase buffer with protease inhibitors and eluted in 15 μL SDS sample buffer.

## Recombinant protein purification and size exclusion chromatography

Hxk2 was cloned into the bacterial expression plasmid pET14b (Novagen) such that the C-terminus contained a 6-histidine tag. Hxk2 expression was induced with 1 mM IPTG at 25°C for 2 hours. Recombinant proteins were purified on Ni-NTA columns (1 ml Ni-NTA Agarose; Qiagen Cat # 30210), washed with 20 ml hexokinase buffer (20 mM HEPES, 5 mM Mg Acetate, 100 mM NaCl, 0.5 mM EDTA, 0.5 mM DTT) with 100 mM imidazole before elution in hexokinase buffer with 500 mM imidazole. Proteins were dialyzed into hexokinase buffer with 5% glycerol and stored at -80°C.

Purified proteins were analyzed by size exclusion chromatography using a TOSOH G2000 $SW_{XL}$ column on a Shimadzu HPLC system. Samples (40 μg protein in 50 μl of buffer) were resolved in hexokinase buffer with or without 2 mM glucose at a 1 mL/min flow rate.

## Fluorescence microscopy

Unless otherwise indicated, imaging was performed using a live-cell microscopy protocol, maintaining yeast in the same medium they were grown in throughout the imaging process. Fluorescent protein localization was performed by growing cells in SC medium with 2% glucose overnight, re-inoculating at an optical density $(OD)_{600}$ of 0.3 into fresh SC medium with 2% glucose, and growing cells until they reached mid-logarithmic phase (an additional 4–5 h) at 30˚C with aeration. For low glucose treatments (SC with 0.05% glucose), cells were washed and incubated as described above in the "Yeast Strains and Growth Conditions" section. For imaging, cells were plated on a 35 mm glass bottom microwell dish coated with 15 μL (0.2 mg/mL) of concanavalin A (MatTek Corporation, Ashland, MA). They were imaged using a Nikon Eclipse Ti2 A1R inverted confocal microscope (Nikon, Chiyoda, Tokyo, Japan) outfitted with a 100 x oil immersion objective (NA 1.49). Images were captured using GaAsP or multi-alkali photomultiplier tube detectors, and the acquisition was controlled using NIS-Elements software (Nikon). All images within an experiment were acquired using identical settings, and images were adjusted evenly and cropped using NIS-Elements.

For S1F Fig, cells were grown as described [11]. 25 μL of cells were loaded onto ConA-coated glass slides. Then the remaining suspension was aspirated off the slide. DAPI (1 μL of 2.5 μg/mL dissolved in 80% glycerol) was added to the cells. Cells were covered with a glass coverslip [11] and then imaged as described above, this time using a Nikon Eclipse Ti2-E inverted microscope (Nikon, Chiyoda, Tokyo, Japan) equipped with an Apo100X objective (NA 1.45) and captured with an Orca Flash 4.0 cMOS (Hamamatsu, Bridgewater, NJ) camera and NIS-elements software (Nikon). These conditions mirror those used in earlier publications of Hxk2-GFP localization [8,11–14].

Fluorescence recovery after photobleaching (FRAP) experiments (see Figs 2F and S2 and S1 and S2 Movies) were performed by adding 25 μL of low-glucose incubated cells to ConA-coated Matek dishes. The experiment was conducted using the confocal microscope described above. First, images were captured before the nuclei were bleached. Next, a region of interest (ROI) for bleaching (see S2F Fig) was defined in the nucelus, using the mScarlet marker reference. A 1 sec pulse of the 488 nm laser (10% power) bleached the nuclei. An image of the bleached cells was captured immediately after and then every minute for 20 mins to monitor nuclear fluorescence recovery.

## Image quantification and statistical analyses

Quantification of nuclear fluorescence and whole-cell fluorescence intensity was done using Nikon General Analysis 3 software (Nikon, Chiyoda, Tokyo, Japan) with the segmentation from NIS-Elements.*ai* (Artificial Intelligence) software (Nikon, Chiyoda, Tokyo, Japan) unless otherwise described. For quantification of whole-cell fluorescence, the NIS.*ai* software was trained on a ground truth set of samples where cells had been manually segmented using the DIC channel images. Next, the NIS.*ai* software was iteratively trained until it achieved a training loss threshold of <0.02, indicating a high degree of agreement between the initial ground truth and the output generated by the software. To measure the mean nuclear fluorescence, we trained the NIS.*ai* software to identify the nucleus using the chromosomally integrated Tpa1-mScarlet nuclear marker (see S2 Table). The NIS.*ai* software was trained using a manually defined ground truth set of nuclear segmentations. Using the General Analysis 3 software, fields of images captured through confocal microscopy were processed so that individual

whole-cell and nuclear objects in a field of view were segmented using the DIC and 561 nm (mScarlet) channels. A parent-child relationship was applied to individual nuclear objects (child) within the same cell (parent) to aggregate them as single objects and pair them to the appropriate whole cell. Any partial cells at the edges of the image were removed along with their child objects. Then the mean fluorescence intensity of each parent or child object was defined in the appropriate channel. All imaging quantification graphs, except for manual quantification data in S1A, S1B, S1D and S1E Fig, were derived using this method.

Manual quantification to measure mean nuclear or whole-cell fluorescence (Fig 1A, 1B, 1D and 1E) was performed using ImageJ software (National Institutes of Health, Bethesda, MD). A 2-pixel wide line was hand drawn around the nuclei using images of the Tpa1-mScarlet nuclear marker to create a mask that was then overlaid on the GFP images, and the mean GFP signal was measured. The same was done to measure whole-cell fluorescence, except lines were drawn around the perimeter of each cell using the GFP or mNG signal since Hxk2 has a diffuse cytosolic distribution. Mean background fluorescence intensity was measured for each image and subtracted from the mean fluorescence measurements to calculate mean nuclear and whole-cell intensities.

Statistical analyses of fluorescence quantification were done using Prism (GraphPad Software, San Diego, CA). Unless otherwise indicated, we performed Kruskal-Wallis statistical tests with Dunn's post hoc correction for multiple comparisons. In all cases, significant p-values from these tests are represented as * p-value<0.1; ** p-value<0.01; *** p-value<0.001; **** p-value<0.0001; not significant (ns) = p-value>0.1. In some instances where multiple comparisons are made, the † or + symbols may be used in place of * to indicate the same p-values but relative to a different reference sample (see the figure legends).

FRAP data were analyzed first by measuring the mean nuclear fluorescence of a nuclear ROI both before and after bleaching. In some cases, nuclei shifted positions at different time points along the lateral plane. In these cases, we manually re-positioned the ROI and re-measured to ensure we did not erroneously measure the cytosolic pool. To account for non-specific photobleaching that occurred due to repeated rounds of imaging, an ROI reference was used in an adjacent cell when no targeted laser bleaching was performed. The mean cytosolic GFP fluorescence in the ROI of the control cell was also measured at each time point. Then the data were normalized using the following equation [104]:

$$ Norm(t) = \frac{Ref_{pre-bleach}}{ref(t)} \times \frac{FRAP_{(t)}}{FRAP_{pre-bleach}} $$

The normalized data were plotted over time, and the recovery rates were calculated by measuring the slope of the linear portion of each recovery plot.

## RNA-Seq sample preparation and analyses

RNA samples were prepared from multiple independent yeast cultures grown on synthetic complete medium using the RNeasy Mini Kit (Qiagen). Sequencing libraries were prepared using the TruSeq Stranded mRNA library method (Illumina). RNA sequences were mapped to *S. cerevisiae* mRNA using the kallisto software package [105]. Each RNA sample yielded 40–50 million reads. mRNA abundance was expressed in transcripts per million mapped reads (tpm). To compare mRNA expression under different conditions, we used a Student's t-test to calculate p values with a false discovery rate threshold of 0.01%. All RNA-seq data have been deposited in the SRA database under accession number PRJNA885127.

### Homology models of the *Sc*Hxk2 dimer

We generated a homology model of the *Sc*Hxk2 dimer using SWISS-MODEL [106–110]. A *Kl*Hxk1 crystallographic dimer (PDB ID 3O1W [41]) served as the template. We removed all alternate locations from the 3O1W PDB file so that each amino acid had only one sidechain conformation. We then copied the two chains, A and B, into two separate files, each containing the respective *Kl*Hxk1 monomer. We separately uploaded these two monomers to the SWISS-MODEL server, together with the full-length sequence of *Sc*Hxk2.

Since the 3O1W template structure covers *Kl*Hxk1 almost entirely—including the critical N-terminal tails—the homology models of each monomer included all *Sc*Hxk2 amino acids except the initial methionine and the terminal alanine. The initial methionine (M1) is cleaved *in vivo* [39,111], so we used UCSF Chimera [112] to add only the C-terminal alanine. To merge the two monomers into a single dimer model, we used multiseq [113], as implemented in VMD [114], to align each monomer to its respective chain in the original 3O1W dimeric structure. Finally, we processed the dimer model with PDB2PQR [115,116], which added hydrogen atoms per the PROPKA algorithm (pH 7.00) [117] and optimized the hydrogen-bond network.

To generate a final model of the *apo* (ligand-free) *Sc*Hxk2 dimer, we subjected the dimer homology model to one round of minimization in Schrödinger Maestro.

To generate a final model of the *holo* (glucose-bound) *Sc*Hxk2 dimer, we aligned a glucose-bound *Kl*Hxk2 dimer (PDB ID 3O5B [41]) to our *Sc*Hxk2 dimer model and copied the aligned glucose molecules. We then added hydrogen atoms to the glucose molecules using Schrödinger Maestro. To resolve minor steric clashes and optimize interactions between the receptor and glucose ligands, we minimized the *Sc*Hxk2/glucose complex using a stepwise protocol. First, we used Schrödinger Maestro to subject all binding-site atoms (excluding the ligand) to one round of minimization. Second, we subjected all protein atoms to two rounds of minimization. Third, we subjected all protein and glucose hydrogen atoms to one round of minimization. Finally, we minimized all the atoms of the complex.

To generate a model of the *holo* (glucose-bound) *Sc*Hxk2 monomer, we simply deleted one of the monomers of our *holo Sc*Hxk2 dimer model. We note that the ATP depicted in Figs 3G and 4A was not part of the model itself (i.e., it was not included in the minimization procedure). We positioned ATP in the active site for visualization by aligning a crystal structure of a homologous protein (6PDT [35]) to each monomer of our *Sc*Hxk2 dimer model.

## Supporting information

**S1 Text. Results, Materials and Methods, and References.**
(DOCX)

**S1 Fig. (accompanies Fig 1). mNG-tagged Hxk2 localizes to the nucleus in response to glucose starvation, and pre-incubation of cells in glycerol before imaging makes interpretations of localization difficult.** (A-B) Manual quantification (using Image J) of the imaging data shown in Fig 1A and comparable to the automated approach used in Fig 1C–1D, to measure (A) mean nuclear fluorescence or (B) the ratio of the mean nuclear/whole-cell fluorescence. Horizontal black lines show the median, and error bars indicate the 95% confidence interval. Dashed yellow and blue lines represent the median value for Hxk2-GFP in high glucose and low glucose, respectively. Black asterisks represent statistical comparisons between low and high glucose for Hxk2-GFP. (C) Confocal microscopy of mNeonGreen-tagged Hxk2 expressed as a chromosomal integration. Co-localization with the nucleus is determined based on overlap with the Tpa1-mScarlet nuclear marker, and a dashed white line indicates the

nucleus. (D-E) Manual quantification (using Image J) or automated quantification (using Nikon.ai and GA3) of the images shown in panel C to measure (D) mean nuclear fluorescence or (E) the ratio of the mean nuclear/whole-cell fluorescence. Horizontal black lines show the median, and error bars indicate the 95% confidence interval. Dashed yellow and blue lines represent the median value for Hxk2-mNG in high glucose and low glucose, respectively. Black asterisks represent statistical comparisons between low and high glucose for Hxk2-mNG. (F) Confocal microscopy of GFP-tagged Hxk2 expressed from a CEN plasmid under the control of the *HXK2* promoter in cells lacking *HXK2* alone. For this experiment, cells were grown in glucose (2% glucose) or shifted into 0.05% glucose (low glucose) for 2 h, but before imaging, these cells were incubated in 80% glycerol with DAPI to recreate the imaging conditions from earlier studies [9–13]. (G) To assess multimerization, we prepared extracts from yeast cells expressing the indicated hexokinase proteins either untagged (-) or tagged (+) with V5 or GFP. Protein inputs were monitored by immunoblotting (bottom two panels). The association of the tagged proteins was assessed by co-immunoprecipitation using anti-V5 beads followed by immunoblotting with anti-GFP (top panel).
(EPS)

**S2 Fig. (accompanies Fig 2). Assessing nuclear dynamics of Hxk2-GFP in response to glucose starvation.** (A) Cells lacking all three hexokinase genes (*hxk1Δ hxk2Δ glk1Δ)* were transformed with empty vector or plasmids expressing wild-type, untagged Hxk2, or Hxk2-GFP. Cell growth ($A_{600}$) in media containing glucose as the carbon source was monitored for 24 hours. (B) Confocal microscopy of GFP-tagged Hxk2 expressed from a CEN plasmid under the control of the *HXK2* promoter in cells lacking *HXK2*. Co-localization with the nucleus is determined based on overlap with the Tpa1-mScarlet nuclear marker, and a dashed white line indicates the nucleus. Cells were imaged at the indicated time points post-shift to glucose-depleted media for up to 24 hours. (C-D) Automated quantification of the images shown in panel B to measure (C) mean nuclear fluorescence or (D) the ratio of the mean nuclear/whole-cell fluorescence. Horizontal black lines show the median, and error bars indicate the 95% confidence interval. Dashed yellow and blue lines represent the median value for Hxk2-GFP in high and 1 hour in low glucose, respectively. Black asterisks represent statistical comparisons between high glucose and low glucose time points. (E) Immunoblot analyses of Hxk2-GFP in whole-cell protein extracts made from cells grown in high glucose or shifted to low glucose for the indicated times. (F) Still frames from FRAP time-lapse images, corresponding to quantified data in Fig 2D, showing pre- and post-bleached cells at selected time points. In the left-hand column, images from *hxk2Δ* cells expressing Hxk2-GFP from a CEN plasmid under the control of the endogenous *HXK2* promoter are shown. In the right-hand column, images from BY4742 cells expressing free GFP from CEN plasmids under the control of the *TEF1* promoter are shown as a control for rapid nuclear recovery after photobleaching. White-dashed circles and arrowheads indicate the location of the nucleus. Yellow circles represent portions of the cytosol in an adjacent cell that was measured and used as a control for the rate of photobleaching due to repeated rounds of imaging over time.
(EPS)

**S3 Fig. (accompanies Fig 3). Hxk2 nuclear localization in response to glucose starvation is also observed in the W303 genetic background. (**A) Confocal microscopy of GFP-tagged Hxk2 or select mutant alleles expressed from a CEN plasmid under the control of the *HXK2* promoter in *hxk2Δ* cells derived from a W303 genetic background. Nuclear localization was determined based on the peri-nuclear ER localization of the mCherry-tagged Scs2-TM construct [14]. This marker was used to generate the white-dashed line that represents the peri-nuclear ER overlay in row 3. (B) A line scan of the region shown as a white line in row 2 of (A)

is provided. The fluorescence intensities for GFP (green line) are graphed based on the left Y-axis values, which in each case measures the mean GFP fluorescence intensity (a.u.) along the line. The fluorescence intensities for the mCherry-tagged peri-nuclear ER (red line) are graphed based on the right Y-axis values, which measure the mean mCherry fluorescence intensity (a.u.) along the line. All fluorescence measures are graphed vs. the distance along the line (microns) from panel A.
(EPS)

**S4 Fig. (accompanies Fig 3). Comparison of the amino acid sequences and structures for *Sc*Hxk1, *Sc*Hxk2, and *Kl*Hxk1.** (A) Clustal Omega amino-acid sequence alignment of *Sc*Hxk1, *Sc*Hxk2, and *Kl*Hxk1. Identical amino acids, conserved residues, and non-conserved residues are shown in the consensus motif as asterisks, colons, or spaces, respectively. Colors represent the amino-acid side-chain functional groups, with red representing alkyl side chains, blue representing acidic side chains, purple representing basic side chains, and green representing neutral side chains. (B-D) Crystal structures of ScHxk1, ScHxk2, and KlHxk1 (PDB IDs 1HKG, 1IG8, and 3O1W) superimposed using UCSF Chimera. The structures are all highly similar, but ScHxk2 and KlHxk1 have the greatest similarity.
(EPS)

**S5 Fig. (accompanies Fig 4). Molecular dynamics simulations demonstrating a possible mechanism of N-terminal-tail dissociation from the enzymatic pocket upon glucose binding.** (A) Early in the simulation, the positively charged V2 terminal amine slides between D417* and D458*, forming salt bridges with both. (B) The bound glucose molecule briefly forms hydrophobic contacts with the V2 side chain. (C) Glucose moves to a different location within the enzymatic cleft, and the V2-D458* salt bridge brakes. At roughly the same time, a hydrogen bond forms between H3 and D86*. (D) The tail beings to dissociate from the opposite-monomer cleft. K7, which previously formed a salt bridge with E457*, now forms a salt bridge with D417* instead. K8, which did not previously interact with the opposite monomer, forms a salt bridge with E457*. (E) Tail dissociation is stabilized by a cation-π interaction between H3 and R423*, and Q10 shifts to form a hydrogen bond with T107*. (F) Finally, the V2 terminal amine forms a salt bridge with E352*, and H3 forms a hydrogen bond with the backbone carbonyl oxygen of N422*.
(EPS)

**S6 Fig. (accompanies Fig 5). A second K/R-rich, putative nuclear localization sequence in Hxk2 does not impact its nuclear partitioning in response to changing glucose conditions.** (A) One monomer of the full-length ScHxk2 homology model is shown in surface representation. Surface-exposed lysine residues are shown in blue, and arginine residues are shown in purple. Labeled amino acids indicate the location of a basic, lysine-rich patch and predicted NLS at residues K54, K59, and K59, as identified by SeqNLS and NLStradamus [15,16]. Note that per PSORTII, Predict NLS, and cNLS Mapper, there are no predicted NLSs in Hxk2 [17–19]. (B) Confocal microscopy of GFP-tagged Hxk2 and K54A, K58A, and K58A mutant expressed from a CEN plasmid under the control of the *HXK2* promoter in cells lacking *HXK2* alone. Co-localization with the nucleus is determined based on overlap with the Tpa1-mScarlet nuclear marker, and a dashed white line indicates the nucleus. (C-D) Automated quantification of the images shown in panel B to measure (C) the mean nuclear fluorescence or (D) the mean nuclear/whole-cell fluorescence ratio. Horizontal black lines show the median, and error bars indicate the 95% confidence interval. Dashed yellow and blue lines represent the median value for Hxk2-GFP in high and low glucose, respectively. Black asterisks represent statistical comparisons between low and high glucose for a specific *HXK2* allele. Blue daggers represent

statistical comparisons between mutant alleles and the corresponding WT Hxk2 in the same medium condition.
(EPS)

**S7 Fig. (accompanies Fig 5). Mutational analyses of lysine-rich putative nuclear localization sequences in Hxk2.** (A) Confocal microscopy of GFP-tagged Hxk2 and K7A, K8A, and K13A mutant expressed from a CEN plasmid under the control of the *HXK2* promoter in cells lacking *HXK2* alone. Co-localization with the nucleus is determined based on overlap with the Tpa1-mScarlet nuclear marker, and a dashed white line indicates the nucleus. (B-C) Automated quantification of the images shown in panel A to measure (B) the mean nuclear fluorescence or (C) the mean nuclear/whole-cell fluorescence ratio. Horizontal black lines show the median, and error bars indicate the 95% confidence interval. Dashed yellow and blue lines represent the median value for Hxk2-GFP in high and low glucose, respectively. Black asterisks represent statistical comparisons between low and high glucose for a specific *HXK2* allele. Blue daggers represent statistical comparisons between mutant alleles and the corresponding WT Hxk2 in the same medium condition. (D) Confocal microscopy of GFP-tagged Hxk2, K13A, or K13R mutants expressed from a CEN plasmid under the control of the *HXK2* promoter in cells lacking *HXK2* alone. Co-localization with the nucleus is determined based on overlap with the Tpa1-mScarlet nuclear marker, and a dashed white line indicates the nucleus. (E-F) Automated quantification of the images shown in panel D to measure € the mean nuclear fluorescence or (F) the mean nuclear/whole-cell fluorescence ratio. Horizontal black lines show the median, and error bars indicate the 95% confidence interval. Dashed yellow and blue lines represent the median value for Hxk2-GFP in high and low glucose, respectively. Black asterisks represent statistical comparisons between low and high glucose for a specific *HXK2* allele. Blue daggers represent statistical comparisons between mutant alleles and the corresponding WT Hxk2 in the same medium condition. (G) Immunoblot analyses of Hxk2-GFP or the K13A mutant from whole-cell protein extracts of WT or tda1Δ cells that were grown in high glucose or shifted to low glucose for 2 h. The full-length Hxk2-GFP and the free GFP breakdown products are shown for each and are equally adjusted so that their abundances can be compared. Total protein stain is shown as a loading control. Note that there is no increase in the free-GFP signal in response to low glucose shift, nor is there increased free GFP from the K13A mutant protein.
(EPS)

**S8 Fig. (accompanies Fig 6). N-terminal mutations in Hxk1 do not alter Hxk1 nuclear propensity.** (A) Confocal microscopy of GFP-tagged Hxk1 and the indicated mutant alleles expressed from CEN plasmids under the control of the *HXK1* promoter in cells lacking *HXK1*. Co-localization with the nucleus is determined based on overlap with the Tpa1-mScarlet nuclear marker, and a dashed white line indicates the nucleus in the GFP with a nuclear outline row. (B-C) Automated quantification of the images shown in panel A to measure (B) the mean nuclear fluorescence or (C) the mean nuclear/whole-cell fluorescence ratio. Horizontal black lines show the median, and error bars indicate the 95% confidence interval. Dashed yellow and blue lines represent the median value for Hxk1-GFP in high and low glucose, respectively. The dashed gray line represents the median value for Hxk1 in high/low glucose media simultaneously for panel C since these values are not distinguishable. Black asterisks represent statistical comparisons between low and high glucose for a specific *HXK1* allele. Blue daggers represent statistical comparisons between mutant alleles and the corresponding WT Hxk1 in the same medium condition.
(EPS)

**S9 Fig. (accompanies Fig 8). Mig1 does not co-purify with Hxk2.** (A) Mig1 does not associate with Hxk2. Extracts were prepared from yeast cells expressing the untagged Hxk2 or Hxk2 tagged with either V5 and Mig1 untagged or tagged with HA as indicated. Protein expression was monitored by western blotting (bottom two panels). The association of Mig1 with Hxk2 was assessed by co-immunoprecipitation using anti-V5 beads, followed by western blotting with anti-HA (top panel).
(EPS)

**S10 Fig. (accompanies Fig 10). Hxk2 nuclear translocation is still regulated in a *snf1*Δ background, independent of S15 phosphorylation status.** (A) Confocal microscopy of GFP-tagged Hxk2 and S15D mutant expressed from a CEN plasmid under the control of the *HXK2* promoter in cells lacking *HXK2* alone or with the additional deletion of *SNF1*. Co-localization with the nucleus is determined based on overlap with the Tpa1-mScarlet nuclear marker, and a dashed white line indicates the nucleus. (B-C) Automated quantification of the images shown in panel A to measure (B) the mean nuclear fluorescence or (C) the mean nuclear/whole-cell fluorescence ratio. Horizontal black lines show the median, and error bars indicate the 95% confidence interval. Dashed yellow and blue lines represent the median value for Hxk2-GFP in high and low glucose, respectively. Black asterisks represent statistical comparisons between low and high glucose for a specific *HXK2* allele. Blue daggers represent statistical comparisons between mutant alleles and the corresponding WT Hxk2 in the same medium condition. Red crosses represent statistical comparisons between WT Hxk2 and Hxk2$^{S15D}$ in the *hxk2*Δ *snf1*Δ background in the same media condition.
(EPS)

**S11 Fig. (accompanies Fig 11). Loss of Hxk2 modestly alters the expression of some glucose-regulated genes, but it does not alter Mig1-induced transcription in response to low glucose.** (A) Analysis of transcript abundances (tpm) for the genes indicated in either WT cells or those lacking HXK2 in high glucose. The genes indicated are selected as they have previously been shown to have some altered expression in *hxk2*Δ cells. We suggest that these modest changes are not due to defective glucose repression but instead are secondary compensation mechanisms for the loss of hexokinase. (B) We took the high-confidence set of genes identified as regulated by Mig1 [20] and then determined which had increased expression (>2-fold) in response to low glucose. We plotted the average tpm values for these genes from WT or hxk2Δ cells grown in high glucose or shifted into low glucose. We find little to no difference in the Mig1-regulation of glucose repression in *hxk2*Δ cells compared to WT cells.
(EPS)

**S1 Table. Yeast strains.**
(DOCX)

**S2 Table. Plasmids.**
(DOCX)

**S3 Table. RNA Seq transcripts per million reads data to accompany Fig 11.**
(XLSX)

**S1 Movie. FRAP of nuclear Hxk2-GFP to accompany Fig 2D.**
(AVI)

**S2 Movie. FRAP of nuclear-free GFP to accompany Fig 2D.**
(AVI)

## Acknowledgments

We acknowledge the support of the Dietrich School of Arts and Sciences Microscopy and Imaging Suite (RRID: SCR_022084). We gratefully acknowledge the support of Dr. Patrick Thibodeau for help with our size-exclusion chromatography assays. We kindly acknowledge the feedback provided by Dr. Jeff Brodsky, his research team, and the Pittsburgh Area Yeast Meeting members before submission. We further acknowledge the support and feedback from members of the O'Donnell lab team.

## Author Contributions

**Conceptualization:** Mitchell A. Lesko, Jacob D. Durrant, Martin C. Schmidt, Allyson F. O'Donnell.

**Data curation:** Mitchell A. Lesko.

**Formal analysis:** Mitchell A. Lesko, Dakshayini G. Chandrashekarappa, Ray W. Bowman II, Chaowei Shang, Jacob D. Durrant, Martin C. Schmidt, Allyson F. O'Donnell.

**Funding acquisition:** Mitchell A. Lesko, Jacob D. Durrant, Martin C. Schmidt, Allyson F. O'Donnell.

**Investigation:** Mitchell A. Lesko, Dakshayini G. Chandrashekarappa, Eric M. Jordahl, Katherine G. Oppenheimer, Jacob D. Durrant, Martin C. Schmidt, Allyson F. O'Donnell.

**Methodology:** Mitchell A. Lesko, Dakshayini G. Chandrashekarappa, Eric M. Jordahl, Ray W. Bowman II, Chaowei Shang, Jacob D. Durrant, Martin C. Schmidt, Allyson F. O'Donnell.

**Project administration:** Jacob D. Durrant, Martin C. Schmidt, Allyson F. O'Donnell.

**Resources:** Martin C. Schmidt.

**Supervision:** Jacob D. Durrant, Martin C. Schmidt, Allyson F. O'Donnell.

**Visualization:** Mitchell A. Lesko, Jacob D. Durrant, Martin C. Schmidt, Allyson F. O'Donnell.

**Writing – original draft:** Mitchell A. Lesko, Jacob D. Durrant, Martin C. Schmidt, Allyson F. O'Donnell.

**Writing – review & editing:** Mitchell A. Lesko, Eric M. Jordahl, Katherine G. Oppenheimer, Ray W. Bowman II, Chaowei Shang, Jacob D. Durrant, Martin C. Schmidt, Allyson F. O'Donnell.

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
