## [Decision Letter · Decision Letter 0]

23 Mar 2023

Dear Dr ODonnell,

Thank you very much for submitting your Research Article entitled 'Hexokinase 2 localizes to the nucleus in response to glucose limitation but does not regulate gene expression' to PLOS Genetics.  The manuscript was fully evaluated at the editorial level and by independent peer reviewers. The reviewers appreciated the attention to an important topic but identified some concerns that we ask you address in a revised manuscript.  We therefore ask you to modify the manuscript according to the review recommendations. Your revisions should address the specific points made by each reviewer.

Please note:

- The formatted review is attached as a pdf (I think - if not I will send it by email).

- The reviewer mentioned "I’m still mystified that the authors find that *HXK2* seems to play no role in glucose repression, even though numerous studies of many individual glucose repressed genes over many (50?) years suggest that it plays a significant role in this process."  Any light you could shed on this issue would be welcome.

Yours sincerely,

Aaron P. Mitchell, PhD

Academic Editor

PLOS Genetics

Gregory Barsh

Editor-in-Chief

PLOS Genetics

Reviewer's Responses to Questions

**Comments to the Authors:**

Reviewer #1: [See also review uploaded as attachmment for saving formatting]

Review of Lesko et al.

The authors have adequately addressed my comments (and, it seems to me, those of the other reviewers).

I’m still mystified that the authors find that HXK2 seems to play no role in glucose repression, even though numerous studies of many individual glucose repressed genes over many (50?) years suggest that it plays a significant role in this process. Nevertheless, I believe this paper will be a worthy contribution to the literature.

A few comments and suggestions the authors may want to address:

Two things confused me. First:

A second layer of glucose-induced regulation must exist for Hxk2 because

739 Hxk2S15D and Hxk2S15A are nuclear excluded even in glucose-grown cells.

“Even in glucose-grown cells” doesn’t make sense to me because Hxk2 is normally nuclear excluded in glucose-grown cells.

Second, In two places in the ms. I’m told that Hxk2 forms dimers (and monomers) in high glucose, but forms predominantly monomers in low glucose:

Rigorous biochemical analyses demonstrate that in glucose-replete conditions, a

89 balance of monomeric and dimeric Hxk2 exists, but Hxk2 shifts to predominantly

90 monomeric when glucose is restricted [29–32].

In a glucose-rich medium, Hxk2 exists in a balance between dimeric and

210 monomeric species [32,33]. Upon glucose starvation, this balance shifts toward the

211 monomeric state [32,33].

But then I’m told that high glucose “destabilizes the dimer” and that “low glucose encourages dimerization”.

In high glucose

346 concentrations, bound glucose might disrupt N-terminal-tail binding within the catalytic

347 pocket to destabilize the dimer (Fig 4A). Alternatively, N-terminal-tail binding may

348 prevent glucose binding in low glucose concentrations, encouraging dimerization.

The ms. is long. I suspect it could be modestly shortened (and made clearer) with aggressive editing. While editing the ms., I urge the authors to follow Strunk and White’s dictum: “avoid needless words.” In the attached file are my suggestions of text that can be deleted (because the necessary formatting doesn't carry over when I paste the text into the website). (I realize this will achieve only modest shortening, but I thought I should provide some examples. And these are only suggestions; I hope they’re useful.)

**Have all data underlying the figures and results presented in the manuscript been provided?**

Reviewer #1: None

PLOS authors have the option to publish the peer review history of their article (what does this mean?). If published, this will include your full peer review and any attached files.

Reviewer #1: No

---

## [Editor Report · Decision Letter 1]

14 Apr 2023

Dear Dr ODonnell,

We are pleased to inform you that your manuscript entitled "Changing course: Glucose starvation drives nuclear accumulation of Hexokinase 2 in S. cerevisiae" has been editorially accepted for publication in PLOS Genetics. Congratulations!

Yours sincerely,

Aaron P. Mitchell, PhD

Academic Editor

PLOS Genetics

Gregory Barsh

Editor-in-Chief

PLOS Genetics

Comments from the reviewers (if applicable):

**Data Deposition**

http://datadryad.org/submit?journalID=pgenetics&manu=PGENETICS-D-23-00104R1

**Press Queries**

---

## [Editor Report · Acceptance letter]

11 May 2023

PGENETICS-D-23-00104R1 

Changing course: Glucose starvation drives nuclear accumulation of Hexokinase 2 in S. cerevisiae 

Dear Dr O'Donnell, 

We are pleased to inform you that your manuscript entitled "Changing course: Glucose starvation drives nuclear accumulation of Hexokinase 2 in S. cerevisiae" has been formally accepted for publication in PLOS Genetics! Your manuscript is now with our production department and you will be notified of the publication date in due course.

With kind regards,

Livia Horvath

PLOS Genetics

On behalf of:
